# Immunomodulatory Properties of Pomegranate Peel Extract in a Model of Human Peripheral Blood Mononuclear Cell Culture

**DOI:** 10.3390/pharmaceutics14061140

**Published:** 2022-05-27

**Authors:** Miodrag Čolić, Marina Bekić, Sergej Tomić, Jelena Đokić, Dušan Radojević, Katarina Šavikin, Nataša Miljuš, Milan Marković, Ranko Škrbić

**Affiliations:** 1Medical Faculty Foča, University of East Sarajevo, 73300 Foča, Bosnia and Herzegovina; 2Serbian Academy of Sciences and Arts, 11000 Belgrade, Serbia; 3Institute for the Application of Nuclear Energy, University of Belgrade, 11000 Belgrade, Serbia; marina.bekic@inep.co.rs (M.B.); sergej.tomic@inep.co.rs (S.T.); milan.markovic@inep.co.rs (M.M.); 4Institute of Molecular Genetics and Genetic Engineering, University of Belgrade, 11000 Belgrade, Serbia; jelena.djokic@imgge.bg.ac.rs (J.Đ.); dradojevic@imgge.bg.ac.rs (D.R.); 5Institute for Medicinal Plant Research “Dr Josif Pančić”, 11000 Belgrade, Serbia; katarina.savikin@gmail.com; 6Faculty of Medicine, University of Banja Luka, 78000 Banja Luka, Bosnia and Herzegovina; natasa.miljus@med.unibl.org (N.M.); ranko.skrbic@med.unibl.org (R.Š.)

**Keywords:** pomegranate, mononuclear cells, cytotoxicity, cytokines, immunomodulation

## Abstract

Pomegranate peel extract (PoPEx) has been shown to have antioxidant and anti-inflammatory properties, but its effect on the adaptive immune system has not been sufficiently investigated. In this study, the treatment of human peripheral blood mononuclear cells (PBMC) with PoPEx (range 6.25–400 µg/mL) resulted in cytotoxicity at concentrations of 100 µg/mL and higher, due to the induction of apoptosis and oxidative stress, whereas autophagy was reduced. At non-cytotoxic concentrations, the opposite effect on these processes was observed simultaneously with the inhibition of PHA-induced PBMC proliferation and a significant decrease in the expression of CD4. PoPEx differently modulated the expression of activation markers (CD69, CD25, ICOS) and PD1 (inhibitory marker), depending on the dose and T-cell subsets. PoPEx (starting from 12.5 µg/mL) suppressed the production of Th1 (IFN-γ), Th17 (IL-17A, IL-17F, and IL-22), Th9 (IL-9), and proinflammatory cytokines (TNF-α and IL-6) in culture supernatants. Lower concentrations upregulated Th2 (IL-5 and IL-13) and Treg (IL-10) responses as well as CD4+CD25hiFoxp3+ cell frequency. Higher concentrations of PoPEx increased the frequency of IL-10- and TGF-β-producing T-cells (much higher in the CD4+ subset). In conclusion, our study suggested for the first time complex immunoregulatory effects of PoPEx on T cells, which could assist in the suppression of chronic inflammatory and autoimmune diseases.

## 1. Introduction

*Punica granatum* (pomegranate) has been used in ancient medicine to treat various diseases and pathological conditions. Due to its beneficial health effect, it has become a focus of current research [1]. The biologically active ingredients are present in the peel, seeds, and arils of the pomegranate. The pomegranate peel, consisting of 26–30% of the total weight of fruit [2], is considered the main medicinal part and contains different minerals, essential amino acids, sugars, unsaturated and saturated fatty acids, tocopherols, sterols, terpenoids, and high concentrations of polyphenols (anthocyanins, flavonoids, ellagitannins, and alkaloids) [1,3,4]. Ellagitannins, the primary polyphenols of pomegranate, consist dominantly of punicalagin, punicalin, ellagic acid, and gallic acid. These compounds possess various protective effects on the human body by neutralizing the destructive effects of free radicals and oxidative/nitrosative stress on the surface of the cell membrane, organelles, and DNA [5] and thus have antimutagenic, antiinflammatory, antifibrosis, anticancer, and anti-aging properties [1,6]. In addition, it has been demonstrated that pomegranate ameliorates hypertension [7], Alzheimer’s disease, diabetes, obesity-related metabolic complications, atherosclerosis, and cardiovascular diseases [1,8,9]. Moreover, a wide spectrum of antibacterial, antiviral, and antiparasitic effects was published [1,10,11,12], including our recent data on SARS-CoV-2 [13].

The most studied effect of pomegranate extract and its polyphenol compounds is related to ameliorating oxidative stress. Oxidative stress is defined as an imbalance between the accumulation and production of oxygen-reactive species (ROS) and the ability of the biological system to detoxify these products that damage lipids, carbohydrates, proteins, and DNA [14]. Polyphenols scavenge free reactive spices of O and N by their OH groups by giving an H group to them [1,10]. Oxidative stress is important in the pathogenesis of various disorders such as cardiovascular diseases, diabetes, chronic inflammation, and malignant tumors. The antioxidant activity of pomegranate varies due to different phenolic contents in their products [7]. In this context, pomegranate juice has been shown to possess the least antioxidant but the most antimutation effects, whereas its methanolic extract showed the most antioxidant but the lowest antimutation activity. The reason for these opposite effects is not clear, but this may be due to variation in the quality and quantity of polyphenols and other bioactive compounds present in different extracts [15,16].

Pomegranate decreases malondialdehyde (MDA) and nitric oxide (NO) levels in different oxidative stress models but increases the levels of reduced glutathione (GSH) content, catalase (CAT), sodium dehydrogenase (SOD), and glutathione peroxidase (GPx) activities [17]. Oxidative stress is tightly associated with apoptosis and other forms of cell death. It has been shown that pomegranate extract induced adenosine triphosphate (ATP) depletion, subG1 accumulation, and apoptosis in oral cancer cell lines but showed no toxicity to normal oral cells [18]. The apoptotic mechanisms involved mitochondrial membrane potential disruption and mitochondrial superoxide generation associated with the differential downregulation of several antioxidant gene mRNA/protein expressions and mitochondrial DNA damage. The process was followed by the downregulation of autophagy [18]. It is known that the autophagy pathway is activated to guarantee the elimination of damaged mitochondria and other damaged organelles to maintain cell survival. In the case of reduced autophagy, cell death could be triggered [19].

Oxidative stress, apoptosis, and autophagy are tightly associated with inflammation. In this context, numerous in vitro and in vivo experiments have shown that the pomegranate extract exhibits a strong anti-inflammatory effect [1,2,12]. The effect was confirmed in models of ulcerous colitis [20,21,22], rheumatoid arthritis [23,24], neuroinflammation [25], skin inflammation [26], ovalbumin-induced asthma in mice [27], lipopolysaccharide (LPS)-stimulated RAW264.7 macrophage cells [28], human neutrophils stimulated with formyl-methionyl-leucyl-phenylalanine (fMLP) and phorbol myristate acetate (PMA) [29]. Similar anti-inflammatory findings were observed in patients with rheumatoid arthritis [30,31] and inflammation in patients with unstable angina or myocardial infarction [32], obese persons [33], or hemodialysis patients [34]. The investigated parameters depended on the goal of the studies, but most of them were associated with decreased production of pro-inflammatory cytokines such as interleukin-6 (IL-6), IL-1β, tumor necrosis factor-α (TNF-α), NO, prostaglandin E2 (PGE2), or polarization of pro-inflammatory M1 macrophages into anti-inflammatory M2 macrophages [1,2,35,36]. The most investigated signaling pathway involved nuclear-factor kappa B (NF-κB) and cyclooxygenase-2 (COX-2) [35,36].

Inflammation is associated with the induction of the immune response, which can be directed in various ways. Some experimental studies showed that pomegranate acts as an immunostimulator [12], which contradicts its strong anti-inflammatory properties. Immunostimulation, as judged by increased production of IgG, was observed in mice following an 8-week supplementation with pomegranate extract [37]. In addition, pomegranate juice feeding in mice resulted in increased phagocytosis [38]. Most studies, mainly related to polyphenols from different sources, demonstrated the immunosuppressive properties of these compounds. The effects depend on the investigated compound, its purity, application dose, and cell culture models [1,2,35,36,39,40,41]. Up to now, no effect of pomegranate peel extract (PoPEx) was investigated in a culture model of human peripheral blood mononuclear cells (PBMC) regarding its immunomodulatory activity. Therefore, this was the primary goal of our study. By using different concentrations of standardized PoPEx we investigated the dose-dependent effects of PoPEx on human PBMC: cytotoxicity, oxidative stress, autophagy, lymphocyte cell surface marker expression, proliferative capability upon stimulation with phytohemagglutinin (PHA), production of pro-and anti-inflammatory cytokines, as well as differentiation of T regulatory cells (Tregs).

## 2. Materials and Methods

### 2.1. Study Protocol

This was an in vitro study performed on human PBMC from healthy volunteers (n = 12), both sexes, aged between 25 and 35 years, who were assigned informed consent after explaining the study objective. The study protocol, which involved the cultivation of PBMC with different concentrations of PoPEx, was approved by the Ethical Committee of the Faculty of Medicine, University of Banja Luka, R. Srpska, Bosnia, and Herzegovina. PoPEx was prepared at the Institute for Medicinal Plant Research, Dr. Josif Pančić, Belgrade, Serbia. The main experiments were performed at the Center for Biomedical Research, Faculty of Medicine, Banja Luka, Institute for the Application of Nuclear Energy, Zemun, Serbia, and the Center for Biomedical Sciences, Medical Faculty Foča, University of East Sarajevo, R. Srpska, Bosnia, and Herzegovina.

### 2.2. Plant Material and Extractions

Pomegranate fruits were collected at a natural locality in the East Herzegovina region (south of Bosnia and Herzegovina). The peel was separated from the fruit manually, dried for 4–6 days at room temperature, and grounded in powder. The powdered pomegranate peel (100 g) was extracted with 50% ethanol, using 1:10 as a solid to solvent ratio. Extraction was performed in an ultrasonic bath at 600 °C for 40 min. After filtration, the extract was evaporated (Büchi R-210 rotary evaporator, Flawil, Switzerland), and the raw material was then used for further extraction and characterization. The total phenolic content in the pomegranate peel extract was analyzed spectrophotometrically using the Folin–Ciocalteu method as previously described [42]. Gallic acid (0–100 mg/L) was used for the preparation of the calibration curve. The results were expressed as mg of gallic acid equivalents per gram of dry weight. The total tannins content was calculated using the method described in the European Pharmacopoeia (Ph Eur 7.0). The percentage content of tannins is expressed as pyrogallol (%, *w*/*w*). The total flavonoid content was measured spectrophotometrically [43], where catechin was used as a standard for the calibration curve. The total flavonoid content of the extract was expressed as mg catechin equivalents per gram of sample. All results represent the mean of three measurements. The detailed protocol is described in our previous paper [44]. The pomegranate peel was deposited in Botanical Garden “Jevremovac” University of Belgrade (voucher specimen No. BEOU 17742).

### 2.3. HPLC Analysis

HPLC analyses were performed on instrument Agilent 1200 RR HPLC (Agilent, Waldbronn, Germany), equipped with a DAD detector, using reverse-phase analytical column Zorbax SB-C18 (Agilent) as we previously described in detail [45]. Mobile phase A was the solution of orthophosphoric acid in water (1% *v*/*v*) whereas mobile phase B was acetonitrile. The flow rate was 1 mL/min. Gradient elution was employed according to the following scheme: 98–90% A (0–5 min); 90% A (5–15 min); 90–85% A (15–20 min); 85–70% A (20–25 min); 70–40% A (25–30 min); 40–0% A (30–34 min). Detection was done at 260 and 320 nm. The quantity of investigated compounds was calculated using calibration curves of authentic standards (punicalagin, punicalin, gallic acid, and ellagic acid). The results are expressed as mg per gram of dry weight. Experiments were repeated three times.

### 2.4. PBMC Cultures

Whole blood samples were collected from healthy volunteers in citrated tubes. PBMCs were isolated by Nycoprep (Nycomed, Oslo, Norway) density-gradient centrifugation and cultured in 96-well flat-bottom plates (Sarstedt, Numbrecht, Germany) at a density of 3 × 10^5^ cells/well in a volume of 200 µL/well). The culture medium was RPMI 1640 (Sigma-Aldrich, St. Louis, MO, USA) supplemented with 10% fetal calf serum and antibiotics (all from Sigma-Aldrich): penicillin (100 units/mL), streptomycin (0.1 mg/mL), and gentamicin 0.08 mg/mL. The cells were incubated at 37 °C in an atmosphere of 5% CO_2_ and were either left in a resting state or treated with a stimulating agent, PHA (10 µg/mL). The unstimulated or PHA-stimulated cultures were treated with doubled, increasing concentrations of PoPEx (6.25–400 µg/mL) and incubated for 4 and 24 h (mRNA expression), 24 h (cytotoxicity, oxidative stress, and autophagy), 3 days (cytokines and phenotype) or 4 days (proliferation). Fresh samples of PoPEx in the culture medium were prepared from the original PoPEx batch and initially dissolved in dimethyl sulfoxide (DMSO). The level of endotoxin in PoPEx at the concentration of 800 μg/mL in the complete culture medium was within the accepted limits (0.92 ng/mL) determined by the Limulus amebocyte lysate (LAL) assay. Cell viability was monitored by Trypan blue exclusion, and the experiments were conducted with at least 95% of initial cell viability.

### 2.5. MTT Assay

PBMC were cultivated in 96-well plates (3 × 10^5^/well; triplicates), in either fresh complete RPMI medium or with different dilutions of PoPEx. After a 24-h incubation period, the plates were centrifuged, and the medium was carefully removed. The solution of 3-[4,5-dimethyl-2-thiazolyl]-2,5-diphenyl tetrazolium bromide (MTT) (Sigma-Aldrich) (100 μL/well, final concentration 100 μg/mL), was added. Wells with different concentrations of PoPEX without cells were used to test the interaction of MTT-developed color with the extract. Wells with an MTT solution without cells served as blank controls. The plates were incubated with MTT for 3 h in an incubator at 37 °C. Dissolution of formazan crystals was done by incubating the MTT-treated cultures with 0.1N HCl/10% SDS (sodium dodecyl sulfate) (100 μL/well) overnight. The next day the optical density (OD) of the developed color was read at 570/650 nm (ELISA reader, Behring II, Marburg, Germany). The results were expressed as the relative metabolic activity compared to the metabolic activity of control cultures. The relative metabolic activity of experimental cultures in relation to control cultures (OD used as 100%) was calculated as follows: metabolic activity (%) = (OD of cultures with PoPEx − OD with PoPEx without cells/OD of control cultures without PoPEx − OD of medium without cells) × 100.

### 2.6. Apoptosis/Necrosis Assay

Apoptosis/necrosis was detected by Annexin-V-FITC/PI) staining kit (R&D, Abingdon, UK), following the manufacturer’s instructions. Briefly, cultivated PBMC were collected and washed with binding buffer, followed by incubation with Annexin-V–FITC and PI. The labeled cells were analyzed on a flow cytometer 8 (LSR II, Becton Dickenson, Franklin Lakes, NJ, USA). Annexin-V^-^FITC^+^ cells were recognized as primary apoptotic cells (early phase of apoptosis), PI^+^ cells were primary necrotic cells, whereas double-positive cells were apoptotic/secondary necrotic cells (late phase of apoptosis).

### 2.7. Quantification of Autophagy by Acridine Orange Staining

Acridine Orange (AO) staining was employed for monitoring acidic vesicular organelles formation related to autophagy. The fluorescent dye AO zinc chloride double salt was purchased from Sigma-Aldrich (St. Louis, MO, USA). PBMCs were stained according to published procedures [18]. Briefly, PBMCs were harvested, washed twice in PBS, and acridine orange was added at a final concentration of 1 µg/mL for the next 30 min at 37 °C. After the incubation period, PBMCs were washed and analyzed on a BD LSR II cytometer by measuring green (510–530 nm) and red (650 nm) fluorescence emissions. To plot the frequency distribution of red and green fluorescence intensity of the single events, data were exported using Diva Software 8.1 (Beckton Dickinson, Franklin Lakes, NJ, USA). The red-to-green ratio of individual events from a representative experiment, after calculation in Excel, was plotted using Prism 8 (GraphPad Software, La Jolla, CA, USA), referring to Thomé et al. study [46].

### 2.8. Quantification of Oxidative Stress

The level of reactive oxygen species (ROS) was evaluated using a 2′,7′-dichlorofluorescein diacetate (DCFDA); St. Louis, MO, USA) staining procedure following the manufacturer’s instructions. Briefly, PBMCs were harvested, washed twice in PBS, and DCFDA was added at a final concentration of 0.5 µg/mL for the next 30 min at 37 °C. After the incubation period, PBMCs were washed and analyzed on a BD LSRII cytometer.

### 2.9. Quantification of PHA-Stimulated PBMCs Proliferation

According to the effects of PoPEx on proliferation capacity, PBMCs were first labeled with CellTrace Far Red dye (Invitrogen, Waltham, MA, USA) to assess the manufacturer’s protocol, followed by PHA stimulation for the next 4 days. After that, the cells were harvested and stained with Propidium Iodide (50 μg/mL, Sigma-Aldrich). CellTrace Far Red dye dilution was analyzed after the exclusion of doublets and PI^+^ cells by BD LSRII cytometer.

### 2.10. Flow Cytometry

The flow cytometry analysis of PHA-stimulated PBMC, co-cultured with PoPEx was performed by staining the cells with directly conjugated antibodies: CD3-PeDazzle, CD4-Alexa Fluor (AF) 700, CD8-phycoerythrin (Pe) Cyanin (Cy)7, CD3-PE, CD69-Allophycocyanin (APC), ICOS-1-APCCy7, CD25-Peridinin-Chlorophyll-protein (PerCP)Cy5.5, IL-10-Fluorescein isothiocyanate (FITC), TGF-β-APC, IL-10-PE, FoxP3-FITC, PD-1-APCCy7 (all from Biolegend Inc., San Diego, CA, USA), IgG1 negative control-PE, IgG1 negative control-FITC, IgG1 negative control-APC, IgG1 negative control PerCPCy5.5 (Thermo Fisher, Waltham, MA, USA), IgG2 negative control APCCy7, IgG1 negative control AF700 (Millipore, Burlington, MA, USA). Surface staining with primary Abs was conducted in PBS/0.1% NaN_3_/0.5% FBS for 30 min at +4 °C, after which the cells were washed twice in PBS/NaN_3_, and the intracellular staining was carried out using a flow cytometry fixation and permeabilization kit (Biolegend, San Diego, CA, USA). Signal overlap between the channels was compensated using single labeled samples before each analysis. Non-specific fluorescence was determined according to isotype control antibodies and fluorescence minus one (FMO) control. At least 5000 cells were analyzed in each sample. Doublets were excluded according to forward scatter (FSC-H/FSC-A), and side scatter (SSC-H/SSC-A) within the sable flow (Time/FSC-A), and dead cells were gated out according to PI staining or low FSC properties.

### 2.11. Real-Time Quantitative PCR

Total RNA was extracted from cultured cells resuspended in 500 µL of TRIzol (Thermo Fisher Scientific) reagent and 100 µL chloroform (Sigma-Aldrich/Merck), followed by centrifugation at 12,000× *g* for 5 min at 4 °C and dissolution of the upper aqueous phase in 600 μL of 70% ethanol. According to the manufacturer’s protocol, the total RNA Purification Mini Spin Kit (Genaxxon Bioscience GmbH, Ulm, Germany) was used for further extraction steps. RevertAid RT Reverse Transcription Kit (Thermo Fisher Scientific, Dreieich, Germany) was used to transcribe 0.5 µg of isolated RNA as a template. The synthesized cDNA was then subjected to Real-Time Quantitative PCR (qPCR) analysis using FastGene 2x IC Green Universal ROX kit according to the manufacturer’s protocol on a 7500 real-time PCR machine (Applied Biosystems, Waltham, MA, USA). The results were normalized against GAPDH for each sample and expressed as a relative target abundance (versus the non-treated sample of each donor) using the 2^−∆∆Ct^ method [47]. Primers used in this study are listed in Table 1. All primers were purchased from Thermo Fisher Scientific.

### 2.12. Cytokine Measurement

The concentrations of TNF-α, IFN-γ, IL-2, IL-4, IL-5, IL-6, IL-9, IL-10, IL-13, IL-17A, IL-17F, and IL-22 were detected simultaneously from a single sample of culture supernatant by a Flow Cytomix Microbeads Assay (BioLegend’s LEGENDplex™) as instructed by the manufacturer. This is a bead-based immunoassay that utilizes the same principles as a sandwich immunoassay but uses the flow cytometry measurement (BD LSR II flow cytometer). The concentration of a particular cytokine was determined using a standard curve, which included positive and negative controls, and LEGENDplex™ data analysis software. All measurements of cytokines were performed in duplicates, then the mean values were calculated.

### 2.13. Statistical Analysis

The normality of the data was tested using the Shapiro–Wilk test. For normally distributed samples unpaired *t*-test with Welch correction was used to statistically compare the differences between experimental and control groups for each parameter measured. Otherwise, the Mann-Whitney test was used for the comparisons. The Friedman test (paired one-way ANOVA) with Dunn’s multiple comparison post-test was used for comparison where more than two groups with paired samples were analyzed, as indicated. The values of *p* < 0.05 were considered to be statistically significant. All analyses were performed with GraphPad Prism 8 (GraphPad Software, La Jolla, CA, USA).

## 3. Results

### 3.1. Characterization of Polyphenols from PoPEX

High-pressure liquid chromatography (HPLC) was used for the characterization of PoPEx. HPLC chromatogram (Figure 1) shows that the main phenolic compounds of PoPEx are punicalagin 67.26 ± 0.81 mg/g, punicalin 31.91 ± 0.22 mg/g, ellagic acid 25.11 ± 0.06 mg/g and gallic acid 9.75 ± 0.05 mg/g.

### 3.2. Dose-Dependent Cytotoxicity of PoPEx in Human PBMC Culture

The first aim of this immunobiological study was to examine whether PoPEx is cytotoxic for human PBMC in culture. PBMC were incubated with double decreasing concentrations of PoPEx, starting from 400 µg/mL till 6.25 µg/mL. Cell viability was assayed by the MTT test. Results presented in Figure 2A show that the metabolic activity of PBMC was significantly decreased at PoPEx concentrations of 50 µg/mL and higher in a dose-dependent manner. The concentration of DMSO in the highest dose of PoPEx was 0.1%, and it had no cytotoxic effect (data not shown). Since PoPEx interferes with color reaction in the test, we used microscopic analysis to study the viability of cells by staining cells with Trypan blue, which marks non-viable cells. As shown in Figure 2B, this method confirmed the results obtained by MTT. After prolonged cell culture for 48 h, more than 75% of cells were dead at the highest concentration (data not shown) and therefore, this concentration was omitted from the next experiments.

To study the mode of cytotoxicity, an Annexin V-Fluorescein Isothiocyanate/Propidium Iodide (Annexin-V-FITC/PI) staining was used. The results presented in Figure 2C,D show that the cytotoxicity in cultures with the two highest concentrations (100 µg/mL and 200 µg/mL) of PoPEx was due to apoptosis. With increasing concentration, the percentage of late apoptotic cells dominated over early phase apoptosis. Minimal necrosis was seen with the highest concentration of PoPEx. The increased apoptosis corresponded to the downregulation of the antiapoptotic BCL-2 molecule at the level of mRNA expression after a 4-h culture with a proapoptotic concentration of PoPEx (100 µg/mL). It is interesting that *BCL-2* expression was upregulated in the presence of lower concentrations of PoPEx (50 µg/mL and 12.5 µg/mL) (Figure 2E).

### 3.3. Modulatory Effect of PoPEx on Autophagy in PBMC Culture

Apoptosis and autophagy are closely related to physiological processes. Based on the findings that PoPEx downregulated autophagy in tumor cell lines [18], we wanted to see whether the same phenomenon is present in PBMC culture. Autophagy, as detected by acridine orange (AO) staining, was decreased in PBMC culture at concentrations of 50 µg/mL and higher (Figure 3A–C). This finding is in accordance with the downregulation of mRNA expression for several molecules involved in the autophagy pathway by using 100 µg/mL of PoPEx after a 4 h culture, such as *BECN1* (coding Beclin 1), Unc-51 like autophagy activating kinase 1 (*ULK1*), GABA type-A receptor-associated protein (*GABARAP*), Ultraviolet irradiation resistance-associated gene (*UVRAG*) and Activating molecule in beclin 1-regulated autophagy protein 1 (*AMBRA1*), whereas no significant changes were observed regarding light chain 3B (*LC3B*), coded by Microtubule-associated proteins 1B light chain 3B (*MAP1LCB*), Sequestosome 1 (*SQSTM1/p62*) and Autophagy gene 5 (*ATG5*) mRNA expression. The expression of *ULK1* was also downregulated in the presence of 50 µg/mL of PoPEx. However, the mRNA expression for other molecules (*BECN1, AMBRA1, p62, ATG5,* and *GABARAP*) was upregulated at 50 µg/mL and/or 12.5 µg/mL. The expression of *UVRAG* and *ATG5* was upregulated in the presence of all three concentrations of PoPEx at 24 h, *GABARAP* and *BECN1* were upregulated at 12.5 µg/mL and 100 µg/mL of PoPEx, whereas *MAP1LCB* was upregulated in the presence of the lowest extract concentration. In contrast, all three concentrations of PoPEx downregulated the expression of *ULK1, AMBRA1*, and *SQSTM1* (Figure 3D).

### 3.4. Modulatory Effect of PoPEx on Oxidative Stress in PBMC Cultures

Autophagy and apoptosis are tightly regulated by oxidative stress. The proapoptotic concentrations of PoPEx (100 µg/mL and 200 µg/mL) induced oxidative stress in PBMC culture after 24 has detected by increased DCFDA fluorescence (Figure 4A,B). This was in accordance with the downregulation of mRNA expression for key regulators of oxidative status in the cells, mitochondrial (MT)-NADH (*MT-ND1*), *MT-ND5*, Nuclear factor-erythroid factor 2-related factor 2 (Nrf2), coded by *NFE2L2* gene, *SOD-1,* and *CAT* after 4 h in the presence of 100 µg/mL of PoPEx. No significant changes were observed in the mRNA expression for Heme-oxygenase 1 (*HMOX1*) and thioredoxin (*TXN*) using this proapoptotic concentration of PoPEx. The expression of several genes (*MT-ND1, MT-ND5, HMOX1, SOD1,* and *TXN*) was upregulated in the presence of non-apoptotic concentrations of PoPEx (50 and/or 12.5 µg/mL), whereas *CAT* was still downregulated. After 24 h, with the exception of the downregulation of *NFE2L2* and *HMOX1*, in the presence of the highest PoPEx concentration (100 µg/mL), the mRNA expression of most genes was upregulated. Lower concentrations downregulated the expression of *CAT* and *HMOX1*, whereas most other genes were upregulated (Figure 4C).

### 3.5. Dose-Dependent Effect of PoPEx on Cell Proliferation and T-Cell Subset Changes in PBMC Culture Stimulated with PHA

PHA was used for stimulation of T-cell proliferation in PBMC cultures. Results presented in Figure 5A,B show a dose dependent-inhibition of cell proliferation from concentrations of 25 µg/mL to 200 µg/mL.

To see whether the inhibition of proliferation affects equally CD4 and CD8 T-cell subsets, a flow-cytometry analysis was performed. As shown in Figure 6A,B, the frequencies of total T-cells (CD3^+^) and CD4^+^ T-cells were reduced at concentrations of 50 µg/mL of PoPEx and higher, whereas the total frequency of CD8^+^ T-cells was not significantly affected (*p* > 0.05), even though a slight downregulation of CD8^+^ T-cells was observed. The decrease of CD4^+^ T-cells was significantly higher than that of CD3^+^ T-cells, which indicated that a significant proportion of viable CD4^+^ T-cells lost membranous CD4 expression.

### 3.6. PoPEx Differently Modulates the Expression of Activation/Inhibitory Molecules on T Cell Subsets

We analyzed the effect of different concentrations of PoPEx on three activation molecules (CD69, CD25, and ICOS) and PD1 (an inhibitory molecule upregulated upon activation) on CD4^+^ and CD8^+^ T-cell subsets in PHA-stimulated PBMC. The results presented in Figure 7 showed a different pattern of expression. CD69, CD25, and ICOS had a higher expression on control CD4^+^ T-cells compared to CD8^+^ T-cells, whereas the opposite was noticed for PD1. PoPEx at the concentrations of 25 µg/mL to 200 µg/mL upregulated the expression of CD69 in a dose-dependent manner. The upregulation of CD69 on CD8^+^ T-cells was seen at the concentration of 12.5–50 µg/mL of PoPEx, followed by its downregulation which was statistically significant at the highest dose (200 µg/mL). Except for the concentration of 50 µg/mL, the expression of CD69 was much higher on CD4^+^ T-cells, as compared to CD8^+^ T-cells.

The expression of CD25 on CD4^+^ T-cells was upregulated at lower concentrations of PoPEx (6.25–25 µg/mL). A slight downregulation was observed at the concentration of 100 µg/mL, followed by a statistically significant increase at the highest concentrations (200 µg/mL). The expression of CD25 was upregulated on CD8^+^ T-cells at lower concentrations of PoPEx (6.25–25 µg/mL). In contrast, higher concentrations (100 µg/mL and 200 µg/mL) were inhibitory.

The expression of ICOS was downregulated on both CD4^+^ and CD8^+^ T-cells at the concentrations of PoPEx from 50–200 µg/mL, dose-dependently. The downregulation of ICOS was more obvious in the CD8^+^ T-cell subset. The expression of PD1 on both T-cell subsets (more pronounced on CD8^+^ T-cells) was upregulated with the two lowest concentrations of PoPEx (6.25 µg/mL and 12.5 µg/mL), followed by its downregulation from 50–200 µg/mL.

### 3.7. PoPEx Differently Modulates Cytokine Production in PHA-Stimulated PBMC Cultures

Twelve cytokines associated with the function of T-cell subsets were analyzed in PHA-stimulated PBMC cultures in different concentrations of PoPEx (Figure 8). The level of TNF-α was significantly decreased at the concentrations of PoPEx from 12.5–200 µg/mL, dose-dependently, whereas IL-6 was downregulated only at higher concentrations (100 µg/mL and 200 µg/mL). Both T helper (Th)1 cytokine (Interferon-γ) (IFN-γ) and Th17 (IL-17A and IL-17F) cytokines were significantly downregulated at all concentrations applied, dose-dependently. Similarly, the production of IL-22 (a Th17/Th22 cytokine) and IL-9 (a Th9 cytokine) was significantly decreased from 12.5–200 µg/mL and 25–200 µg/mL, respectively. The dynamics of Th2 cytokines were different. Whereas the level of all three cytokines (IL-4, IL-5, and IL-13) was decreased at concentrations of 50–200 µg/mL (IL-4 and IL-5) or 100 µg/mL and 200 µg/mL (IL-13), the levels of IL-5 and IL-13 were increased at lower concentrations of PoPEx (12.5 µg/mL or 25 µg/mL, respectively). The level of IL-10, a key Treg cytokine, was increased at lower concentrations of PoPEx (12.5 µg/mL) and significantly decreased at 50–200 µg/mL of PoPEx. Of all these cytokines, the level of IL-2 was upregulated at 25 µg/mL and 50 µg/mL. In all other cultures, including the control, its level was very low.

### 3.8. Modulatory Effect of PoPEx on Tregs

The expression of key immunoregulatory cytokines, IL-10, and transforming growth factor-β (TGF-β) was analyzed within CD4^+^ and CD8^+^ T- cell subsets. As shown in Figure 9A,B, the expression of IL-10 was upregulated on both subsets in the presence of PoPEx, starting from 12.5 µg/mL. The highest expression was seen at 50 µg/mL. The expression of TGF-β was upregulated from 50 µg/mL to 200 µg/mL (CD4^+^ T-cells) or 25–200 µg/mL (CD8^+^ T-cells). The levels were highest at 200 µg/mL of PoPEx, especially in CD4^+^ T-cells.

When the frequency of Tregs was analyzed within CD4^+^ T-cells (CD4^+^CD25^hi^Foxp3^+^) (Figure 10A,B), its proportion was higher in the presence of a lower concentration of PoPEx (12.5 µg/mL). Although the proportions of IL-10^+^ Tregs and TGF-β^+^ Tregs were very low, both in control and PoPEx-treated cultures, there was a statistically significant relative increase of IL-10^+^ Tregs at higher concentrations of PoPEx in total T-cells (CD3^+^) and both CD4^+^ and CD8^+^ T-cell subsets. The frequency was higher in the CD4^+^ T-cell subset. The proportion of TGF-β^+^ Tregs in control cultures was higher than IL-10^+^ Tregs, but a similar pattern of TGF-β^+^ Tregs frequency was seen in the presence of PoPEx.

## 4. Discussion

Many studies published so far showed a wide range of biological and clinically relevant activities of pomegranate on cancer, degenerative, metabolic, and chronic inflammatory diseases, thus making this fruit and its polyphenolic compounds (especially ellagitannins) as very important factors for health promotion and medical application [48,49,50].

Of many effects, two important mechanisms stand out: the inhibition of tumor growth due to the induction of oxidative stress and apoptosis without a significant impact on normal cells; strong anti-inflammatory potential. The immune system is of crucial importance in defense against infectious agents and tumors, and in this context, published data mainly support the immunosuppressive properties of ellagitannins. Since immunosuppression and antitumor immune response are opposite effects, a research imperative is to better elucidate pomegranate-associated immunomodulation due to the scarcity of such data. This was the reason why we used a well-proven model of PHA-stimulated PBMC culture to screen the dose-dependent effect PoPEx on immune parameters dominantly associated with T-cell functions.

The first part of the results was related to the cytotoxicity of PoPEx. We showed that concentrations of PoPEx higher than 50 µg/mL are cytotoxic for about 25% (100 µg/mL) and 40% (200 µg/mL) of PBMC, due to apoptosis, which contradict previous results about the relative resistance of normal cells to apoptosis by PoPEx or its polyphenols, in contrast to the sensitivity of cancer cells [51]. The concentration of 50 µg/mL could be considered as sub cytotoxic because a slight decrease in cell viability (about 10%) was negligible. In our experiments, we confirmed that higher concentrations of PoPEx induced apoptosis of PBMC equally in lymphocyte and monocyte subsets (data not shown) and that apoptosis was accompanied by increased oxidative stress and reduced autophagy. The opposite effects were observed with lower, non-cytotoxic, concentrations of PoPEx. In addition, decreased expression of mRNA for antiapoptotic *BCL-2* molecule was observed.

Up until now, most studies investigated the proapoptotic activity of pomegranate constituents on tumor cell lines, whereas their effects on immune cells are relatively scarce. Previous research has shown that punicalagin, a main constituent of PoPEx, induced apoptosis in prostate and colon cancer cell lines, promyelocytic leukemia cells, and glioma cells by increasing the expression of the proapoptotic marker (Bax) X, downregulating the expression of antiapoptotic markers (Bcl-XL and Bcl-2) or suppressing nuclear factor (NF)-*k*B, depending on the investigated mechanisms [51]. In addition, punicalagin induced apoptosis in NB4 and MOLT-4 cells via intrinsic and extrinsic apoptotic pathways by activating caspase-3/-8-9 and altering Bax/Bcl-2.

We showed oxidative stress in PBMC cultures in the presence of higher concentrations of PoPEx based on increased DCFDA fluorescence and downregulation of mitochondrial DNA genes (*MT-ND1* and *MT-ND5* isoforms) involved in the mitochondrial respiratory chain [52] and mRNA for antioxidative enzymes such as *CAT, SOD1,* and *NFE2L2* after 4 h. These changes were mostly reversed and/or upregulated after 24 h. CAT decomposes hydrogen peroxide into water and oxygen, whereas Nrf2 is a critical transcription factor that regulates the basal and stress-inducible activation of many cytoprotective genes, including glutathione and *TXN* [53]. PoPEx has been shown to exert both pro-oxidant and antioxidant activity depending on the applied concentrations and cell lines, similar to what we demonstrated in PBMC cultures. For example, low doses of PoPEx (2.5 µg/mL to 40 µg/mL) inhibited the UVB-induced oxidative stress in keratinocyte HaCaT cells. In contrast, high doses of the extract (100 µg/mL to 200 µg/mL) triggered oxidative stress in several types of cancer cell lines such as lung cancer, leukemia, and fibrosarcoma [54,55,56]. Recent data showed that higher concentrations of PoPEx (100 µg/mL) triggered mitochondrial membrane potential (MitoMP) disruption and mitochondrial superoxide (MitoSOX) generation associated with the differential downregulation of several antioxidant gene mRNA/protein expressions in oral cancer cells [18].

In contrast, the antioxidant activity of pomegranate and its polyphenols has been shown both in vitro and in vivo. The in vitro effect was demonstrated in a model of linoleic acid oxidation [57]. The antioxidant activity of PoPEx phytocompounds in vivo was mediated via different mechanisms, such as the inhibition of lipid peroxidation, scavenging or neutralizing ROS, activating or inhibiting several signaling pathways, and modulating gene expression [58]. Tugcu et al., 2017, demonstrated that the activity of GSH and GSH-Px was increased following the administration of pomegranate juice, while no statistically significant difference was found for SOD among the treated and untreated groups [59]. We showed that lower, non-cytotoxic concentrations of PoPEx, induced expression of most genes involved in the antioxidative activity. However, their dynamics were different in the early (4 h) and late (24 h) periods, most probably due to different patterns of particular antioxidative mechanisms.

Our results demonstrated that pro-oxidative and proapoptotic concentrations of PoPEx are accompanied by decreased autophagy, as revealed by decreased AO fluorescence. Almost identical results were obtained with AO staining using three oral carcinoma cell lines treated with PoPEx [18]. However, we extended our study to the expression of different autophagy-related genes. Cao et al., 2019, also showed that punicalagin at concentrations of 100 µM to 400 µM decreased the viability of RAW264.7 macrophages and inhibited their autophagy at lower concentrations (25 µM and 50 µM) after stimulation with LPS by inhibiting the FoxO3a autophagy signaling pathway [60]. It is well-known that autophagy is essential for both the survival and apoptosis of various cells and that the final fate of a cell depends on the type of cells and its biology. ULK1 in the complex with other autophagy-related proteins is important for the initiation of autophagy [61,62]. Beclin1 is involved in the initiation of autophagosome formation [63], whereas UVRAG is a component of Complex II which has a role in the later stages of autophagy [64]. Atg5 promotes the autophagosome fusion with lysosomes and helps the conjugation of LC3 with phosphatidylethanolamine (PE). In addition, Atg5 activates the extrinsic apoptosis pathway and is responsible for cross-talk among different forms of cell death [65]. The Atg8 family proteins (LC3/GABARAP) are critical for autophagosome-lysosome fusion, whereas AMBRA1 is a regulator of autophagy and apoptosis [66]. p62 is an autophagy substrate used as a reporter of autophagy activity by delivering ubiquitinated cargoes for autophagic degradation [67]. We showed that except for *MAP1LC3B*, *SQSTM1/p62,* and *ATG5*, which were not significantly modulated, all other autophagy-related genes were downregulated in the presence of 100 µg/mL of PoPEx. In contrast, except for downregulated *ULK1*, all genes were upregulated with lower concentrations of the extract in the early period of cultivation (4 h).

Apoptosis and autophagy can be synergistic or opposite processes because a complex interplay between them exists at different signaling levels. For example, punicalagin exerts the cytotoxic effect by suppressing proliferation and promoting apoptosis and autophagy by activating the caspase cascade, altering Bax and Bcl-2, and regulating autophagy via mTOR/ULK1 signaling in human NB4 and MOLT-4 leukemia cell lines [68]. Punicalagin induces apoptotic and autophagic cell death in human U87MG glioma cells [69]. Ganesan et al., 2020, identified 35 different apoptosis/autophagy-related proteins that have been modulated by punicalagin in a colorectal cell line [70]. Autophagy promotes cell survival through activating Beclin1-A in the endoplasmic reticulum (ER) and can switch to apoptosis when caspases in the mitochondria are activated. The Bcl2 and AMBRA1 in both the ER and mitochondria act as important regulators of autophagy and apoptosis [66]. However, several lines of evidence suggest that autophagy inactivation by apoptosis is a crucial event for cell death execution [71].

The interplay between oxidative stress and autophagy is also well-documented and depends on whether the oxidative stress is physiological or excessive (pathological) [72]. The mitochondrial (m) ROS-induced oxidative stress provokes autophagy to protect the cells from apoptosis, and antioxidants attenuate autophagic activation [73]. In contrast, excessive ROS production incurs the accumulation of oxidative stress, which impairs or dysregulates autophagy and can lead to cell death. On the other hand, impairment or dysregulation of the autophagic process causes mitochondrial dysfunction, increasing mROS production [74]. This concept aligns with our findings, although the alterations in autophagy gene expression may not correlate with protein levels. Therefore, such results need to be explored in future experiments because, at the moment, this was not the primary aim of the study.

The anti-inflammatory effects of pomegranate and its constituents have been demonstrated in various experimental models and human diseases [28,74,75]. In a summary of a recent review, Baradaran Rahimi et al., 2020, emphasized that pomegranate and ellagic acid modulate the inflammatory process by reducing many inflammatory cytokines and chemokines such as TNF-α, IL-1β, IL-6, IL-8, and IL-18. The target signaling molecules, receptors, or transcription factors include COX-2, NF-κB, Janus kinase (JNK), inducible nitric oxide synthase (iNOS), P38, extracellular signal-regulated kinase (ERK), Nrf2, peroxisome proliferator-activated receptor (PPAR)-α, and PPAR-γ [1]. We tested the modulatory effect of PoPEx on only two proinflammatory cytokines from the innate cytokine family (TNF-α and IL-6) in PHA-stimulated PBMC cultures and confirmed previous findings. TNF-α was inhibited by both lower and higher PoPEx concentrations, whereas inhibition of IL-6 was seen only at higher doses. The differences could be due to the fact that under certain conditions, IL-6 may also have an anti-inflammatory role [76].

The dominant part of our results was related to the effect of PoPEx on immunomodulation with a particular effect on T-cell proliferation, phenotypic changes, and T-cell producing cytokines. We showed a prominent, dose-dependent, antiproliferative effect of PoPEx on lymphocytes in PHA-stimulated PBMC cultures. The results are in accordance with those published by Mastrogiovanni et al., 2019, with bovine PBMC [77]. In our study, the effect seen with higher proapoptotic concentrations (100 µg/mL and 200 µg/mL) of PoPEx was partly due to the induction of apoptosis and, in this context, the response of CD4 and CD8 T-cells was different. However, non-cytotoxic concentrations (25 µg/mL and 50 µg/mL) were also inhibitory. CD4^+^ T-cells were significantly more affected, not only due to apoptosis but also due to decreased expression of the CD4 molecule. Although this unusual phenomenon deserves to be elucidated in the next experiments, this is not a new finding because, for example, certain antioxidants such as MnTBAP cause a rapid and reversible downregulation of CD4 on T cells in vitro and in vivo. This effect, which resulted from the internalization of membrane CD4 T-cell molecules into clathrin-coated pits, was followed by disruption of the CD4/p56Lck complex. Such a finding may be relevant for impaired CD4^+^ T-cell activation [78].

PHA is a mitogen for human T-cells [79]. Within seconds, a complex cascade of signal transduction is initiated upon T-cell activation and several cell surface markers are upregulated, each at a different stage of the activation process. The earliest activation marker is CD69, which is an inducible cell surface glycoprotein expressed upon activation via the TCR or the IL-2 receptor (CD25). It plays a role in the proliferation and survival of activated T lymphocytes. [80]. CD69 regulates the differentiation of regulatory T (Treg) cells as well as the secretion of IFN-γ, IL-17, and IL-22. Its ligand is Galectin-1 (Gal-1) [81]. CD25 is the alpha chain of the trimeric IL-2 receptor and is considered the most prominent cellular activation marker. It is expressed constitutively on the surface of several subsets of peripheral blood lymphocytes, such as regulatory and resting memory T-cells. The appearance of CD69 on the plasma membrane of activated cells is faster than that of CD25 [81]. ICOS has been implicated in various immune outcomes. It is involved in the induction and regulation of Th1, Th2, and Th17 immunity and dictates the course of both cellular and adaptive humoral immunity [82]. PD-1 is a member of the CD28 family, which modulates T-cell function and is primarily upregulated on CD4 and CD8 T-cells surface upon activation by antigen-specific and non-specific stimulation. Like the inhibitory CTLA-4 receptor, PD-1 may be upregulated early to potentially prime negative regulatory feedback mechanisms to limit the immune response and inflammation [83].

We showed that the inhibited proliferation of T cells in higher concentrations of PoPEx was followed by decreased expression of all four (CD69, CD25, ICOS, and PD1) investigated molecules. This was best visible on CD8^+^ T-cells with 100 µg/mL and 200 µg/mL of the extract. However, increased expression of these molecules on both T-cell subsets with the two lowest concentrations of PoPEx (6.25 µg/mL and 12.5 µg/mL), without increased proliferation, suggests that PoPEx differently affects signaling molecules involved in T-cell proliferation and the expression of activation markers. This hypothesis is supported by similar findings (inhibited proliferation but upregulated expression of activation molecules) when 25 µg/mL and 50 µg/mL of PoPEx were used. Several publications showed that polyphenols target a number of molecules involved in T cell proliferation and activation [84]. Punicalagin inhibited the activation of the nuclear factor of activated T-cells (NFAT), downregulated the expression of interleukin-2 from anti-CD3/anti-CD28-stimulated murine splenic CD4^+^ T-cells, and suppressed mixed leukocytes’ reaction (MLR) without exhibiting cytotoxicity to these cells. It is not clear why the expression of activating markers (CD69 and CD25) was increased, and ICOS decreased on CD4^+^ T lymphocytes at the highest concentration of PoPEx. One possibility, which deserves to be tested, could be related to their expression by a specific CD4^+^ T subset, resistant to the cytotoxic effect of high doses of PoPEx. This was the only analyzed CD4^+^ T-cell subset that could have different functional properties compared with other CD4- T helper cells. CD4+ T helper cells differentiate into T helper (Th) 1, Th2, Th9, Th17, Th22, Th follicular (Thf), and Tregs, depending on the antigen stimulation [45,85,86,87]. Our results showed that PoPEx exerted a strong immunomodulatory effect on T cells in PBMC cultures which is dominantly characterized by a prominent downregulation of Th1, Th17, and Th9 cytokine production. There are many examples in the literature that dietary polyphenols are immunosuppressive [36], but no such complex immunoregulatory potential was published for PoPEx. In one study, lower concentrations of ellagic acid decreased IL-1β, IL-6, IL-1Ra, and IL-10 production by PBMC incubated with a tumor cell line (HT-29), whereas higher concentrations inhibited the production of all cytokines examined, except IFN-γ [88]. In another study, ellagic acid increased expression of Regulated upon Activation, Normal T-cell Expressed and Presumably Secreted (RANTES), IL-2, and IL-1β by primary human gingival epithelial cells. In contrast, the expression of IL-6, IL-8, and TNF-α was decreased [89]. Lu et al., 2020, showed that PoPEx had a positive preventive and therapeutic effect on experimental autoimmune encephalomyelitis (EAE) in mice by inhibiting the infiltration of peripheral inflammatory cells into CNS with a decrease in the proportion of CD4^+^IL-17^+^ and CD4^+^IFN-γ^+^ cells. [90]. Similarly, Stojanović et al., 2017, demonstrated in EAE and type 1 diabetes that PoPEx alleviated symptoms of the diseases, which was accompanied by inhibition of immune cell infiltration into pancreatic islets and decreased IL-17 and IFN-γ production in gut-associated lymphoid tissue in vivo and in vitro [44]. Lu et al., 2020, showed that PoPEx ameliorated the severity of EAE via modulation of gut microbiota [90]. A new formulation of PoPEx significantly ameliorated EAE [91]. In addition, a nanoformulation of pomegranate seed oil improved cognitive disability in patients with multiple sclerosis [92]. The possible beneficial effects of PoPEx in autoimmunity, as suggested in our in vitro study, maybe also relevant for other in vivo studies showing that pomegranate juice reduced fecal calprotectin levels in inflammatory bowel disease patients with a high risk of clinical relapse [93]. Moreover, punicalagin ameliorated experimental lupus nephritis by inhibiting protease-activated receptor-2 [94]. The treatment with propolis, pomegranate, and grape pomace extracts alleviated the severity of the clinical symptoms of RA, which was followed by a decrease in serum levels of IL-17, IL-1β, and IL-17–triggering cytokines [95].

Anderson and Teuber, 2010, showed that walnut kernel polyphenolics and purified ellagic acid inhibited human PBMC proliferation stimulated with PHA, α-CD3, and phorbol myristate acetate (PMA)/ionomycin, while simultaneously increasing the IL-2 production. IL-13 and TNF-α production decreased, while no change was observed in IL-4 production [96]. We found that the Th2 response (as judged by increased levels of IL-5 and IL-13, but not IL-4) is enhanced in the presence of lower concentrations of PoPEx, but at higher concentrations, all cytokines were downregulated. IL-2 is upregulated only in the presence of 25 µg/mL and 50 µg/mL of PoPEx despite inhibited proliferation. IL-2 acts in an autocrine manner by binding its receptor on T cells, thus facilitating cell cycle progression [78]. The increase of IL-2 production simultaneously with upregulated CD25 expression in the absence of proliferation in our study indicates that the polyphenols from PoPEx can render T cells unresponsive to their IL-2 production. In addition, polyphenols can bind to IL-2, its receptor, or interfere with specific molecules responsible for IL-2 signaling, which could be explored in future studies. Another possible mode of antiproliferative action of PoPEx could be extrapolated from ellagic acid experiments, which showed that tannins, as selective protein kinase C (PKC) inhibitors, suppress the PKC activity [97] or induce the cyclin-dependent kinase (Cdk) inhibitory protein p21 and G1 arrest in cancer cell lines [98].

Upregulation of the Th2 response in the presence of PoPEx, a new phenomenon, which has not been described so far, could be due to inhibited Th1 and Th17 responses because they are mutually antagonistic with Th2 [85,99]. The relevance of this pathway deserves to be further explored due to the complexity of Th2 cytokines. Although Th2 cytokines are critical for humoral immunity, antihelmintic response, and allergy, they are mainly anti-inflammatory [85]. For example, IL-13 inhibits proinflammatory cytokines, chemokines, and profibrogenic cytokines synthesis by blocking NF-κB and JNK/AP-1 activation [100]. In addition, the cytokine was able to protect BALB/c mice from autoimmune myocarditis by regulating macrophage differentiation [101].

A new finding in our study was related to the effect of PoPEx on Tregs. We observed an increase in the proportion of CD4^+^CD25^hi^Foxp3^+^ cells in PBMC cultures treated with lower concentrations of PoPEx which correlated with increased production of IL-10. Our findings are similar to the results of Lu et al., 2020 [90], who showed that the treatment of mice with EAE with PoPEx was followed by an increased proportion of CD4^+^Foxp3^+^ and CD4^+^IL-10^+^ in the spleens and brain infiltrates. Tregs are phenotypically and functionally heterogeneous cell populations. Human Tregs produce IL-10 and TGF-β, two main pleiotropic immunosuppressive cytokines which play key roles in protecting the host from infection-associated immunopathology, autoimmunity, and allergy [102,103]. Although the effect of PoPEx was not investigated in the context of immunosuppressive and antitumor effects of IL-10 and TGF-β, it has been shown that ellagic acid inhibited the growth of breast cancer cell lines via the TGF-β/Smads pathway [104] or inhibited inflammation by increasing IL-10 production [105]. We observed an increase in the proportion of both IL-10^+^ and TGF-β^+^ cells within both CD4^+^ and CD8^+^ T-cell subsets in the presence of high concentrations of PoPEx, which contradicts decreased levels of IL-10 in supernatants of such cultures. There are at least two possible explanations for such a discrepancy. The first explanation could be related to decreased total numbers of Tregs in cultures with high concentrations of PoPEx. The second hypothesis could be associated with increased utilization of Treg cytokines which are required for the downregulation of pro-inflammatory Th1 and Th17 responses [99]. However, these hypotheses deserve to be tested in future experiments. Some other immunosuppressive cytokines, such as IL-35 [106] and IL-37 [107], are relevant to be studied in the complex immunoregulatory roles of PoPEx and its ellagitannins.

Although our study showed for the first time the complexity of the immunomodulatory activity of PoPEx, translation of these data from in vitro studies to in vivo models and clinical trials requires a detailed analysis of components from PoPEx and their bioactivity. Our current study showed that all three main components (punicalagin, punicalin, and ellagic acid) act synergistically to modulate the immune response. This will be an advantage in understanding the molecular mechanisms involved in these processes.

## 5. Conclusions

Our results show that PoPEx exerts a dose-dependent effect on human PBMC in culture. While higher concentrations induce oxidative stress, apoptosis, and inhibited autophagy, non-cytotoxic concentrations of the extract possess significant immunomodulatory activity on PBMC upon stimulation with PHA. It is manifested by inhibition of cell proliferation, a marked decrease of CD4 expression on T cells, suppressed Th1, Th17, Th9, and pro-inflammatory cytokines but augmented Th2 and Tregs responses together with the increased CD4^+^CD25^hi^Foxp3^+^ cell frequency. Non-cytotoxic concentrations of the extract increased the expression of autophagy-related, antioxidative, and *BCL-2* genes and the inhibitory PD-1 marker. These results contradict the upregulated expression of CD69, CD25, and IL-2, indicating the complexity of the PoPEx action on the immune cell functions.

## Figures and Tables

**Figure 1 pharmaceutics-14-01140-f001:**
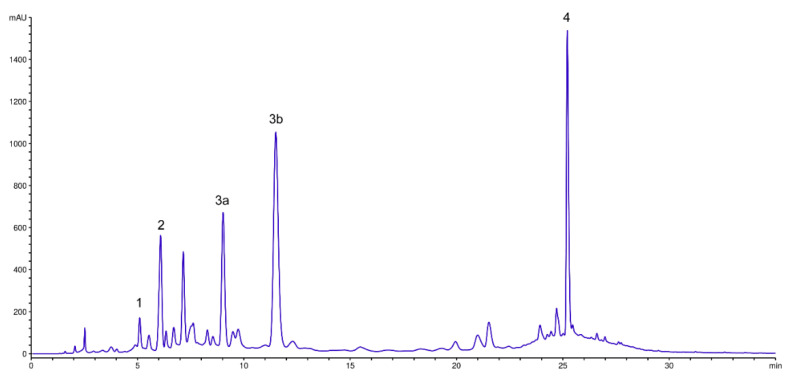
HPLC of PoPEx. 1. Gallic acic; 2. Punicalin; 3a. Punicalagin-α; 3b. Punicalagin-β; 4. Ellagic acid.

**Figure 2 pharmaceutics-14-01140-f002:**
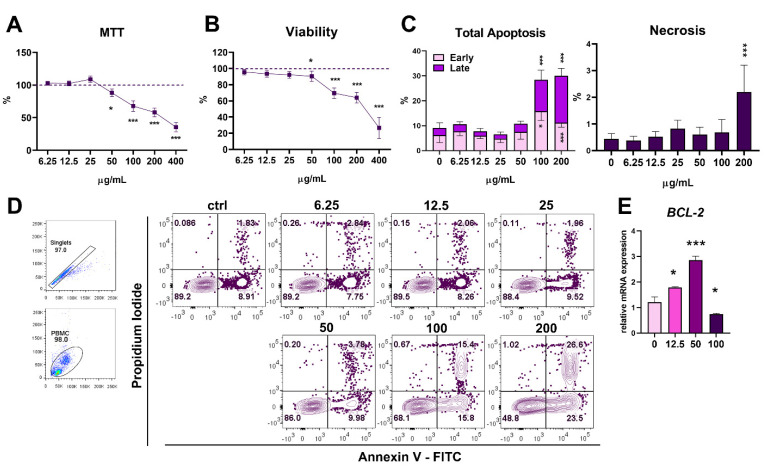
Cytotoxicity of PoPEx in culture with PBMC. Cytotoxicity of PBMC was determined in the culture of PBMC (3 × 10^5^/well) treated with increasing doses of PoPEx (6.25–400 µg/mL) for 24h, followed by the analysis of (**A**) relative metabolic activity by MTT assay; (**B**) relative viability, as determined by Trypan blue exclusion assay; (**C**) The summarized data on the percentage of apoptotic cells (Annexin-V+ PI- for early apoptotic cells; Annexin-V + PI+ for late apoptotic cells) and necrotic (Annexin-V-PI+) is shown in, as determined by Annexin-V/PI staining and flow cytometry analysis (a representative analysis is shown in (**D**)). (**E**) Relative *BCL-2* mRNA expression in PBMC treated with 12.5 µg/mL, 50 µg/mL, or 100 µg/mL for 24 h, was determined by qPCR using the 2^−∆∆Ct^ method and β-actin as a reference gene in the treated groups. The summarized data (**A**–**C**) is shown as mean ± SD from five independent experiments, whereas data in (**E**) is shown as mean ± SD from three independent experiments. * *p* < 0.05, *** *p* < 0.005, compared to control non-treated PBMC (Friedman test; Dunn’s multiple comparison post-test).

**Figure 3 pharmaceutics-14-01140-f003:**
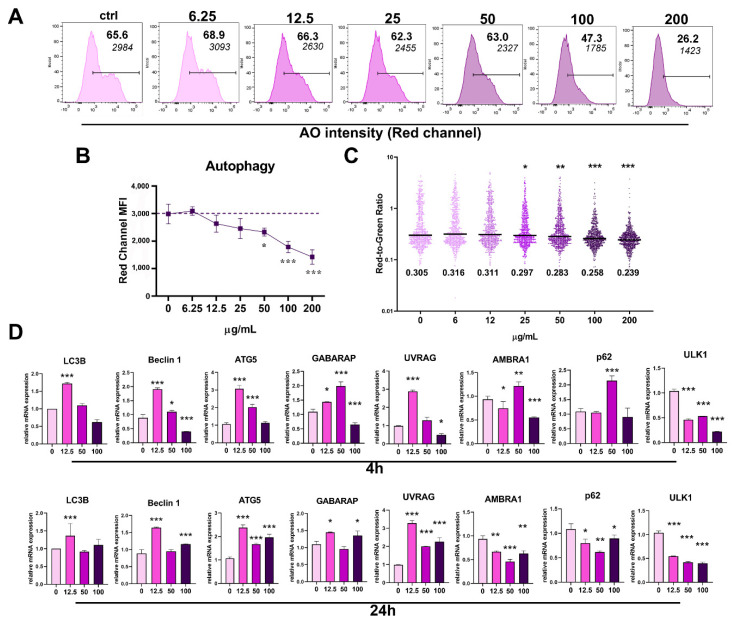
The effects of PoPEx on autophagy in PBMC. Autophagy in PBMC was determined after the cultures of PBMC (3 × 10^5^/well) were treated with increasing doses of PoPEx (6.25–200 µg/mL) for 24h, by acridine orange (AO) staining and flow cytometry analysis. (**A**) A representative analysis of red fluorescence AO intensity is shown, along with the summarized data (**B**). (**C**) The red to green fluorescence ratio was determined by dividing FL2 channel MFI with FL1 channel MFI in AO-stained samples (wo compensation) for each gated event in FCS files. (**D**) Relative mRNA expression for indicated autophagy-regulatory genes in PBMC treated with 12.5 µg/mL, 50 µg/mL, or 100 µg/mL for 4 h or 24 h, was determined by qPCR using the 2^−∆∆Ct^ method, and β-actin as a reference gene in the treated groups. The summarized data (**B**) are shown as mean ± SD from four independent experiments, whereas data in (**D**) is shown as mean ± SD from three independent experiments. * *p* < 0.05, ** *p* < 0.01, *** *p* < 0.005, compared to control non-treated PBMC (Friedman test; Dunn’s multiple comparison post-test).

**Figure 4 pharmaceutics-14-01140-f004:**
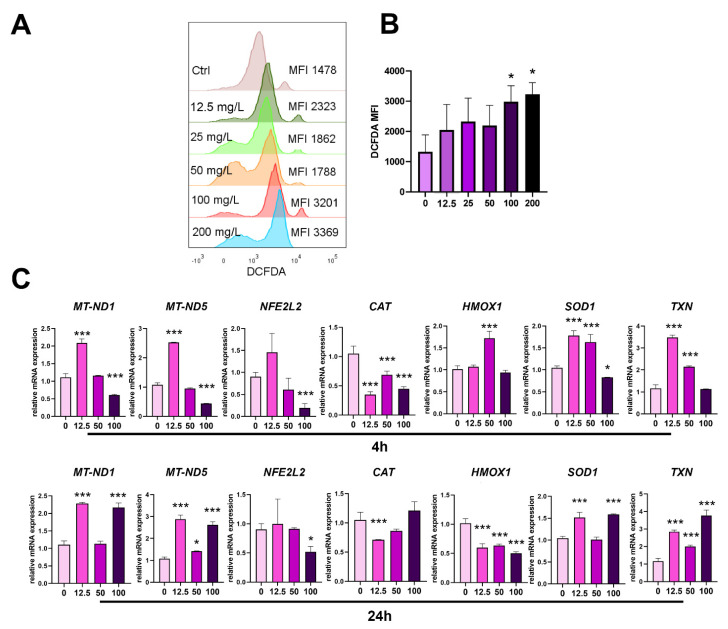
The effects of PoPEx on oxidative stress in PBMC. PBMC (3 × 10^5^/well) were cultured with PoPEx (6.25–200 µg/mL) for 24 h, followed by DCFDA staining and analysis by flow cytometry. (**A**) A representative analysis of DCFDA intensity is shown, along with the summarized data (**B**). (**C**) Relative mRNA expression for indicated genes in PBMC treated with 12.5 µg/mL, 50 µg/mL or 100 µg/mL for 4 h or 24 h, was determined by qPCR using the 2^−∆∆Ct^ method and β-actin as a reference gene in the treated groups. The summarized data (**B**,**C**) are shown as mean ± SD from three independent experiments. * *p* < 0.05, *** *p* < 0.005, compared to control non-treated PBMC (Friedman test; Dunn’s multiple comparison post-test).

**Figure 5 pharmaceutics-14-01140-f005:**
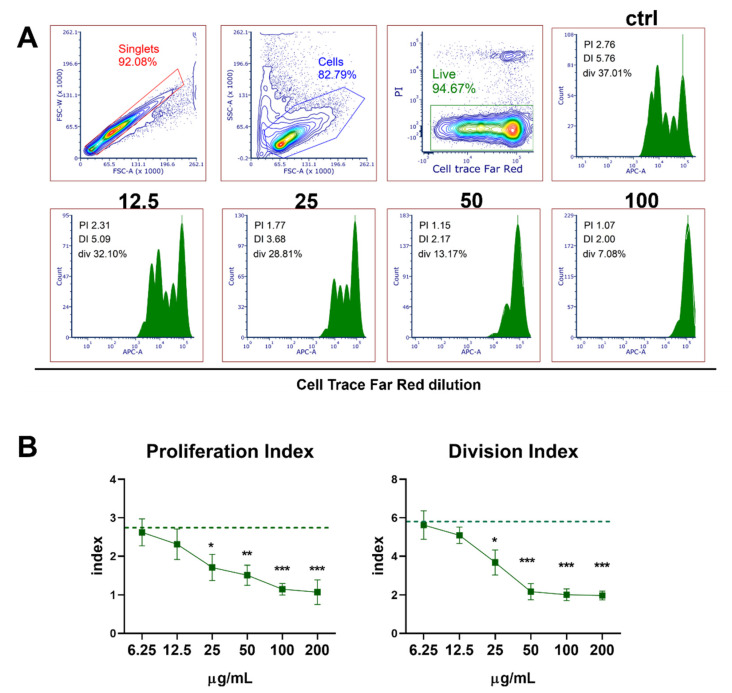
The effects of PoPEx on PHA-stimulated proliferation of PBMC. PBMC (3 × 10^5^/well) pre-labeled with CellTrace Far-Red were cultured with PoPEx (6.25–200 µg/mL) or without PoPEx (ctrl) in the presence of PHA (10 µg/mL) for 4 days, followed by the analysis of Far-Red dilution by flow cytometry. (**A**) A representative gating strategy (the exclusion of doublets, cell debris, and dead (PI+ cells) is shown, and the analysis of Far-Red dilution is shown on histograms (PI-proliferation index, DI-division index, or div-percentage of division) from one experiment, out of 5 with similar results. (**B**) The summarized data for proliferation index and division index is shown as mean ± SD from 5 independent experiments * *p* < 0.05, ** *p* < 0.01, *** *p* < 0.005, compared to control non-treated PHA-PBMC (Friedman test; Dunn’s multiple comparison post-test).

**Figure 6 pharmaceutics-14-01140-f006:**
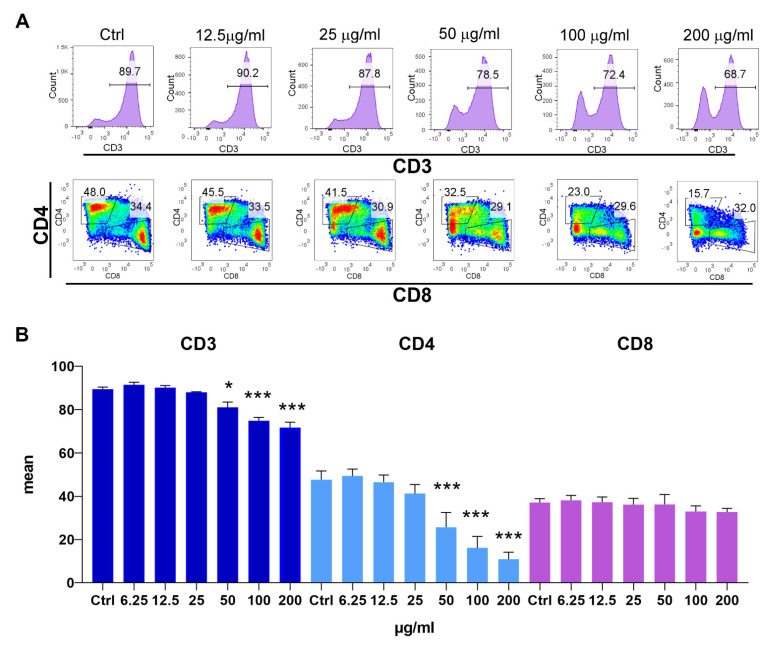
The effects of PoPEx on CD3, CD4 and CD8 expression in PHA-stimulated PBMC. PBMC (3 × 10^5^/well) pre-labeled with CellTrace Far-Red were cultured with PoPEx (6.25–200 µg/mL) or without PoPEx (ctrl), and PHA (10 µg/mL) for 3 days, followed by the analysis of CD3, CD4 and CD8 expression within gated lymphocytes (excluded doublets and cell debris). (**A**) A representative data on CD3 expression within lymphocytes (upper row) and CD4/CD8 co-expression (within 100% CD3^+^ cells) (lower row) is shown, and (**B**) the summarized data is shown as mean ± SD from three independent experiments * *p* < 0.05, *** *p* < 0.005, compared to control non-treated PHA-PBMC (Friedman test; Dunn’s multiple comparison post-test).

**Figure 7 pharmaceutics-14-01140-f007:**
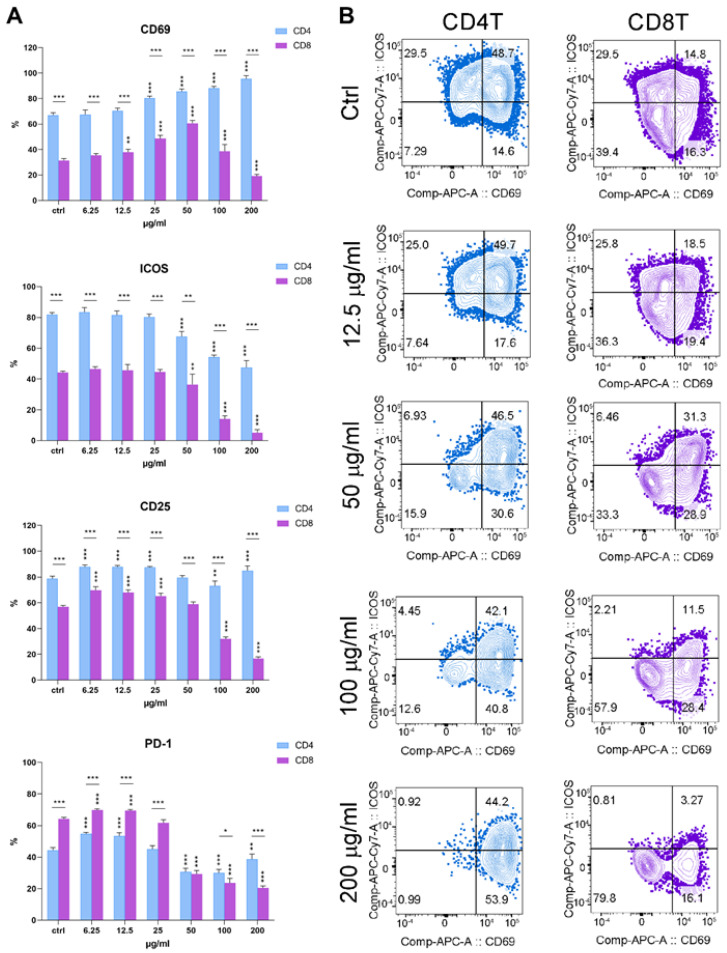
The effects of PoPEx on the expression of activation markers in PHA-stimulated PBMC. PBMC (3 × 10^5^/well) pre-labeled with CellTrace Far-Red were cultured with PoPEx (6.25–200 µg/mL) or without PoPEx (ctrl), and PHA (10 µg/mL) for 3 days, followed by the analysis of CD69, ICOS-1, CD25 and PD-1 on CD3^+^CD4^+^ (CD4 T) or CD3^+^CD8^+^ (CD8 T) T-cells. (**A**) Summarized data is shown as mean % ± SD (n = 3) of marker-positive cells and (**B**) the representative plots from one experiment, out of three with similar results, is shown. * *p* < 0.05, ** *p* < 0.01, *** *p* < 0.005, compared to control non-treated PHA-PBMC or as indicated by line (One-way ANOVA; Dunn’s multiple comparison post-test).

**Figure 8 pharmaceutics-14-01140-f008:**
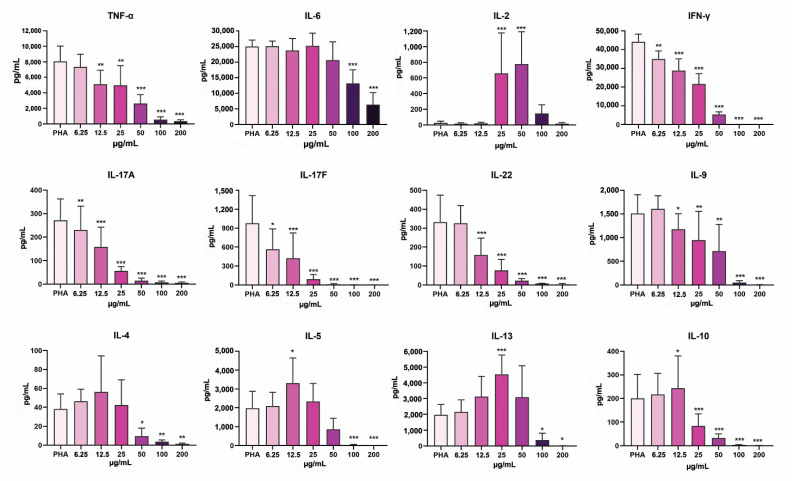
The effects of PoPEx on the production of cytokines by PHA-stimulated PBMC. PBMC (3 × 10^5^/well) were cultured with PoPEx (6.25–200 µg/mL) or without PPoPEx (PHA) in the presence of PHA (10 µg/mL) for 3 days, followed by a collection of cell-culture supernatants and measurements of indicated cytokine levels by flow cytometry, as described in Materials and Methods. The data is shown as mean (pg/mL) ± SD from 8 independent experiments. * *p* < 0.05, ** *p* < 0.01, *** *p* < 0.005, compared to control non-treated PHA-PBMC (Friedman test; Dunn’s multiple comparison post-test).

**Figure 9 pharmaceutics-14-01140-f009:**
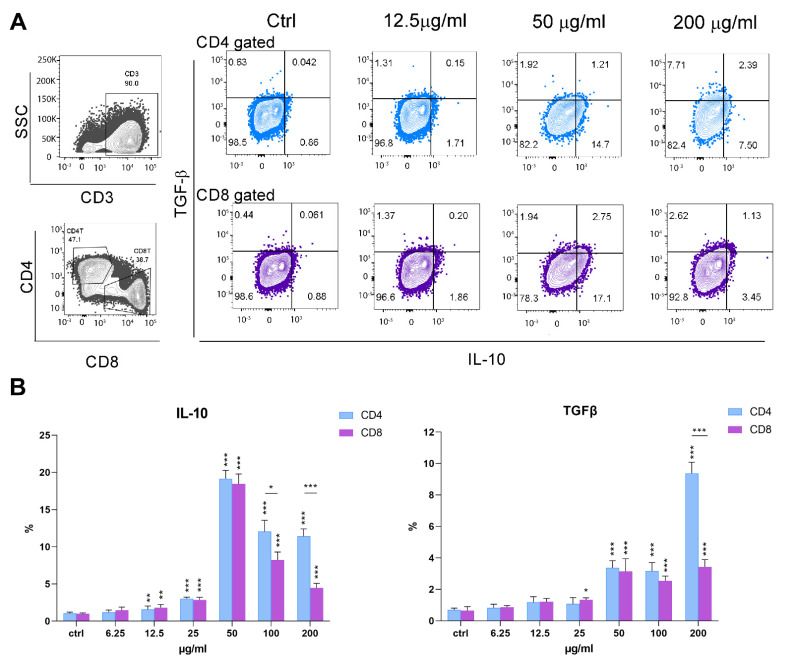
The effects of PoPEx on the expression of IL-10 and TGF-β in PHA-stimulated PBMC. PBMC (3 × 10^5^/well) were culured with PoPEx (6.25–200 µg/mL) or without PoPEx (ctrl), in the presence of PHA (10 µg/mL) for 3 days. (**A**) A representative analysis of IL-10 and TGF-β is shown, as determined within CD3^+^CD4^+^ (light blue) or CD3^+^CD8^+^ T-cells (purple), and (**B**) the summarized data is shown as mean% ± SD (n = 3). * *p* < 0.05, ** *p* < 0.01, *** *p* < 0.005, compared to control non-treated PHA-PBMC or as indicated by line (One-way ANOVA; Dunn’s multiple comparison post-test).

**Figure 10 pharmaceutics-14-01140-f010:**
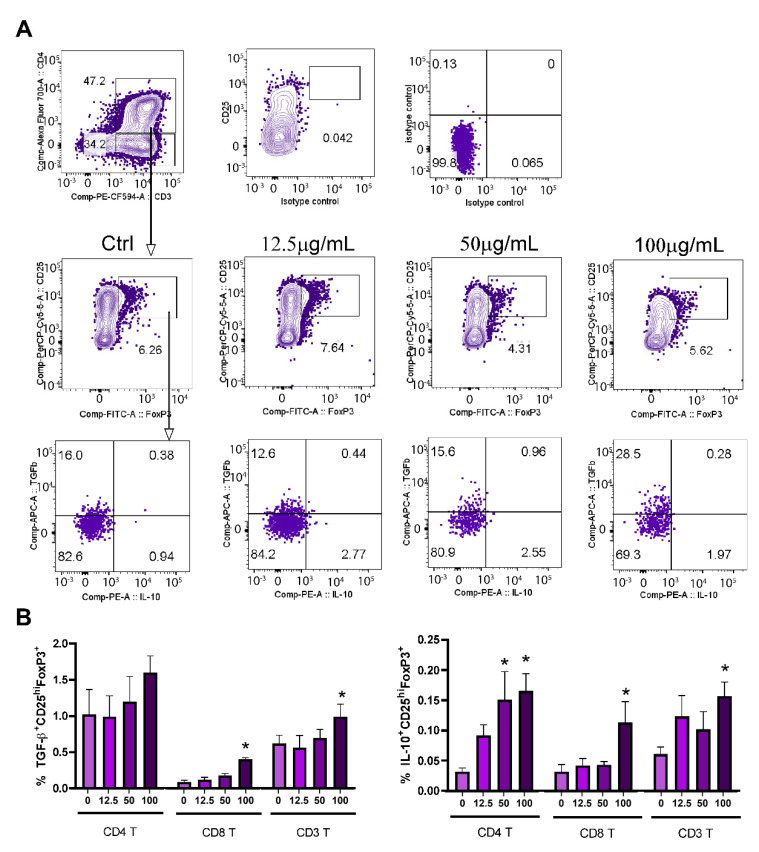
The effects of PoPEx on CD4^+^ and CD8^+^ Tregs within PHA-stimulated PBMC. PBMC (3 × 10^5^/well) were cultured with PoPEx (12.5 µg/mL, 50 µg/mL or 100 µg/mL) or without PoPEx (ctrl), in the presence of PHA (10 µg/mL) for 3 days. (**A**) A representative analysis of IL-10 and TGF-β expression within CD25^hi^FoxP3^+^CD3^+^CD4^+^ (CD4 T) is shown, along with isotype and CD25/isotype (FMO) control. Similar analysis was performed within CD25^hi^FoxP3^+^ CD3^+^CD4^−^ (CD8 T) and for total CD25^hi^FoxP3^+^CD3^+^ (CD3 T) cells (not shown). (**B**) The summarized data on % of TGF-β^+^CD25^hi^FoxP3^+^ and IL-10^+^CD25^hi^FoxP3^+^ within CD3 T, CD4 T and CD8 T are shown as mean ± SD (n = 3). * *p* < 0.05, compared to control non-treated PHA-PBMC (One-way ANOVA; Dunn’s multiple comparison post-test).

**Table 1 pharmaceutics-14-01140-t001:** List of primers used in this study.

Gene Name	Primer Sequence 5′–3′
*GAPDH_F*	GTGAAGGTCGGAGTCAACG
*GAPDH_R*	TGAGGTCAATGAAGGGGTC
*ATG5_F*	CACAAGCAACTCTGGATGGGATTG
*ATG5_R*	GCAGCCAC GGACGAAACAG
*MAP1LC3B_F*	TTCAGGTTCACAAAACCCGC
*MAP1LC3B_R*	TCTCACACAGCCCGTTTACC
*BECN1_F*	CTGGGACAACAAGTTTGACCAT
*BECN1_R*	GCTCCTCAGAGTTAAACTGGGTT
*SQSTM1_F*	GCCAGAGGAACAGATGGAGT
*SQSTM1_R*	TCCGATTCTG GCATCTGTAG
*UVRAG_F*	AGGAAGGAGTGCACTGCAAA
*UVRAG_R*	AGGCAACTTGACACCGCATA
*ULK1_F*	TTTTGTTTCTCCGTTGGGGC
*ULK1_R*	ACTCTTCCCGGGCTGCTAAT
*AMBRA1_F*	GGTGGGAGGAGAGGGGATAG
*AMBRA1_R*	CGAGGGGCATGTCATCATTT
*GABARAP_F*	CCCTCGTCCCGCTGATTTTA
*GABARAP_R*	ATCCCTCCAGCTTGTACCCA
*MT-ND1_F*	CCTCCTACTCCTCATTGTACCCATTC
*MT-ND1_R*	GAGTGTGCCTGCAAAGATGGTAGAG
*MT-ND5_F*	GTTTCATCCTCGCCTTAGCATGA
*MT-ND5_R*	AGTCAGGGGTGGAGACCTAATTGG
*TXN_F*	GAAGCAGATCGAGAGCAAGACTG
*TXN_R*	GCTCCAGAAAATTCACCCACCT
*CAT_F*	AGTGATCGGGGGATTCCAGA
*CAT_R*	AAGTCTCGCCGCATCTTCAA
*NFE2L2 _F*	AGGTTGCCCACATTCCCAAA
*NFE2L2 _R*	AGTGACTGAAACGTAGCCGA
*HMOX1_F*	CTCCCAGGGCCATGAACTTT
*HMOX1_R*	GGGAAGATGCCATAGGCTCC
*SOD1_F*	ACAAAGATGGTGTGGCCGAT
*SOD1_R*	AACGACTTCCAGCGTTTCCT
*BCL2_F*	TCGCCCTGTGGATGACTGA
*BCL2_R*	CAGAGACAGCCAGGAGAAATC

## Data Availability

All data are included in this article.

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
