# Peer review of "Immunomodulatory Properties of Pomegranate Peel Extract in a Model of Human Peripheral Blood Mononuclear Cell Culture"

_pharmaceutics, 2022, doi:10.3390/pharmaceutics14061140_

Round 1

Reviewer 1 Report

In this study, the treatment of human peripheral blood mononuclear cells (PBMC) with pomegranate peel extract (PoPEx) resulted in cytotoxicity at concentrations of higher than 100µg/mL due to the induction of apoptosis and oxidative stress, whereas autophagy was reduced. At the first attempt, pomegranate peel extract was investigated in a culture model of human mononuclear cells, regarding its immunomodulatory activity. It was an in vitro study performed on human PBMC from healthy volunteers (n=12), both sexes, aged between 25 and 35 years who were assigned informed consent after an explanation of the study objective. Authors performed MTT assay, Apoptosis/necrosis assay, Quantification of autophagy by Acridine Orange staining, Quantification of oxidative stress, Quantification of PHA-stimulated PBMCs proliferation, Flow cytometry, Real-Time Quantitative PCR, Cytokine measurement, Statistical analysis. The main phenolic compounds in extract are punicalagin 67.26 ± 0.81 mg/g, punicalin 31.91 ± 0.22 mg/g, ellagic acid 25.11 ±  0.06 mg/g and gallic acid 9.75 ± 0.05 mg/g.

The cytotoxicity in cultures with the two highest concentrations (100 and 200 µg/mL) of PoPEx was due to apoptosis. With increasing concentration, the percentage of late apoptotic cells dominated over early phase apoptosis. Autophagy, as detected by acridine orange (AO) staining, was decreased in PBMC culture at concentrations of 50 µg/mL and higher.

The pro-apoptotic concentrations of PoPEx induced oxidative stress in PBMC culture after 24 hours as detected by increased Dichlorodihydrofluorescein diacetate (DCFDA) fluorescence.

Other significant results are present for proposed assays. The dominant part o results are related to the effect of PoPEx on immunomodulation with a particular effect on T- cell proliferation, phenotypic changes, and T-cell producing cytokines. The authors showed a prominent, dose-dependent, anti-proliferative effect of PoPEx on lymphocytes in PHA-stimulated PBMC cultures. PoPEx differently modulated the expression of activation markers (CD69, CD25, ICOS) and an inhibitory marker (PD1), depending on the dose and T- cell subsets. PoPEx (12.50µg/mL and higher) suppressed the production of Th1 (IFN- γ), Th17 (IL-17A, IL-17F, and IL-22), Th9 (IL-9), and proinflammatory cytokines (TNF-α and IL-6) in culture supernatants. Lower concentrations up-regulated Th2 (IL-5 and IL-13) and Treg (IL-10) responses as well as CD4+CD25hiFoxp3+ cell frequency. In conclusion, the study is well written, very complex, and with interesting results.

Author Response

We thank the reviewer for kind opinion, we much appreciate it.

Reviewer 2 Report

This is an interesting and complete in vitro paper investigating the effects  of the pomegranate peel extract in a model of human peripheral blood mononuclear cell culture.The data demonstrate complex and often dicothomic dosedependent effects of PoPEX in this in vitro system and related oucomes.

In  fact, in vitro culture of  human peripheral blood mononuclear cells (PBMC) PoPEx (range 6.25µg/mL- 400µg/mL) exerted cytotoxicity at concentrations of 100µg/mL and higher, due to the induction of apoptosis and oxidative stress, whereas it reduced autophagy . 
On the contrary, non-cytotoxic concentrations exhibited the opposite effect, but inhibited PHA-induced PBMC proliferation and significantly decreased the expression of CD4. PoPEx differently modulated the expression of activation markers (CD69, CD25, ICOS) and an inhibitory marker (PD1), depending on the dose and T- cell subsets. At concentrations of 12.50µg/mL and higher PoPEX suppressed the production of Th1 (IFN- γ), Th17 (IL-17A, IL-17F, and IL-22), Th9 (IL-9), and proinflammatory cytokines (TNF-α and IL-6) in culture supernatants. Lower concentrations up-regulated Th2 (IL-5 and IL-13) and Treg (IL-10) responses as well as CD4+CD25hiFoxp3+ cell frequency. Higher concentrations of PoPEx increased the frequency of IL-10- and TGF-β-producing T cells (much higher in the CD4+ subset). 

While I agree that these data are of interest,I think that ,also due to the several experimental settings considered, the message of the paper is unclear and difficult to follow and it could be improved.

  1. What is the concentration of LPS in the PoPEX preparation they have used? Have they used a single batch from one preparation or different batches ? The concentration of LPS should be known
  2. The paper is too long , especially in the discussion section and should be reduced. 
  3. The Authors should discuss the limitation of this , though elegant , study in the translation in the cliunical setting.One of this is that PoPEX is a mixture with variables concantrations of components and this renders difficult the reproduction of data. Also, development of nutraceutical products is rendered more difficutl from lack of IP  protection. Have the Authors any suggestion to overcome these difficulties ? Do they envisage one main component that can be principally responsible for the observed effects ? Some literature studies that I quote below addrees these points.
  4. I would suggest the Authors to refer to concentrations of PoPEX observed in other in vitro studies in cancer cell line, RAW cells or others to confirm and or discuss the observed effects
  5. I understand that certain doses of PoPEX could be detrimental for autoimmune diseases. Hence translation of these studies in in vivo models of autoimmune diseases would be necessary. This should be commented . Appropriate, preclinical and even clinical studies hsould be quoted and discussed . Vallarino et al., Molecules 2022.  Petrou, Mult Scler. Relat Disord 2021. Lu , Gut Microbes 2020. Seo Y, Int J. Mol. Sciences, 2020. Parisi V, Molecules 2020. Scaioli |E., Trials, 2019 etc.
  6. The Authors should quote and discuss known seminal in vivo data on antiinflammatory effects of IL13 and IL10  Nicoletti F et al., 1997. Howard M, J. Exp. Med. 1993 as well as the relevance of TGF beta in certain autoimmune diseases such as multiple sclerosis Nicoletti F. Clin Exp.Immunol., 1998. Sotriropoulus MG, 2020. It is of interest as indicated above that preclinical data indicate a benefial effect of a new formualtion of pomegranate peel extract in EAE.
  7. The Authors should quote and discuss the relevance of future studies investigating the role of PoPEX on other pro and anti inflammatory cytokines implicated in autoimmune diseases such as MIF, IL35, IL37 . Gunther S. Drugs Discovery Today 2019 etc

Author Response

  1. What is the concentration of LPS in the PoPEX preparation they have used? Have they used a single batch from one preparation or different batches ? The concentration of LPS should be known

Answer: The concentrations of endotoxin in PoPEx at the concentrations of 800 µg/mL in the complete culture medium was 0.92 ng/mL (a double higher dose than used in the experiments), as determined by a Limulus amebocyte lysate (LAL) assay (added, see lines 172-174).

  1. The paper is too long , especially in the discussion section and should be reduced. 

Answer: We reduced some parts of the Discussion. However, it was difficult to keep the reduced longevity of the text due to a number of requirements addressed in the next points. So we tried to keep the answers below as short as possible. Certain suggested references were not accepted or used similar

  1. The Authors should discuss the limitation of this , though elegant , study in the translation in the cliunical setting.One of this is that PoPEX is a mixture with variables concantrations of components and this renders difficult the reproduction of data. Also, development of nutraceutical products is rendered more difficutl from lack of IP  protection. Have the Authors any suggestion to overcome these difficulties ? Do they envisage one main component that can be principally responsible for the observed effects ? Some literature studies that I quote below addrees these points.

Answer: The response Is added in the last paragraph of the discussion (lines 816-822)

  1. I would suggest the Authors to refer to concentrations of PoPEX observed in other in vitro studies in cancer cell line, RAW cells or others to confirm and or discuss the observed effects

Answer: This has been already discussed “For example, low doses of PoPEx (2.5 µg/mL to 40 µg/mL) inhibited the UVB-induced oxidative stress in keratinocyte HaCaT cells. In contrast, high doses of the extract (100 µg/mL to 200 µg/mL) triggered oxidative stress in several types of cancer cell lines such as lung cancer, leukemia, and fibrosarcoma (see lines 570-574). We added some other concentrations from the cited literature (see lines 575, 595-597 )

  1. I understand that certain doses of PoPEX could be detrimental for autoimmune diseases. Hence translation of these studies in in vivo models of autoimmune diseases would be necessary. This should be commented . Appropriate, preclinical and even clinical studies hsould be quoted and discussed . Vallarino et al., Molecules 2022.  Petrou, Mult Scler. Relat Disord 2021. Lu , Gut Microbes 2020. Seo Y, Int J. Mol. Sciences, 2020. Parisi V, Molecules 2020. Scaioli |E., Trials, 2019 etc.

Answer: The suggested studies (although not quite necessary in our opinion) are briefly commented throughout the text. Unfortunately, this addition extended the length of the manuscript (see lines 751-762).

  1. The Authors should quote and discuss known seminal in vivo data on antiinflammatory effects of IL13 and IL10  Nicoletti F et al., 1997. Howard M, J. Exp. Med. 1993 as well as the relevance of TGF beta in certain autoimmune diseases such as multiple sclerosis Nicoletti F. Clin Exp.Immunol., 1998. Sotriropoulus MG, 2020. It is of interest as indicated above that preclinical data indicate a benefial effect of a new formualtion of pomegranate peel extract in EAE.

Answer: Some of the suggested references are commented on. Some are not quite relevant, so other ones are incorporated (see lines 784-790).

7. The Authors should quote and discuss the relevance of future studies investigating the role of PoPEX on other pro and anti inflammatory cytokines implicated in autoimmune diseases such as MIF, IL35, IL37 . Gunther S. Drugs Discovery Today 2019 etc

Answer: MIF is not incorporated, because there are so many other pro-inflammatory cytokines that are not investigated. The references for immunoregulatory cytokines IL-35 and IL-37, are added (lines 813-815)

Reviewer 3 Report

The subject of the manuscript Immunomodulatory Properties of the Pomegranate Peel Extract in a Model of Human Peripheral Blood Mononuclear Cell Cul- ture is novelty and very interesting.

The experimental data from the present manuscript suggest for the first time complex immunoregulatory effects of Pomegranate peel extract (PoPEx) on T cells, which could make it beneficial in the suppression of chronic inflammatory and autoimmune diseases.

The text is clear and easy to read.

The manuscript is well written and has a significant contribution to the field.

The design research is well described.

The results are clearly presented.

The literature consulted is varied and relatively recently.

At the reference list, some names of species are not italic.

Example: references number 3, 21, 25, 26, 31, 36, 46 – Punica granatum

Please to correct!

Author Response

Example: references number 3, 21, 25, 26, 31, 36, 46 – Punica granatum

Answer: Corrected

Reviewer 4 Report

The concept of phenotypic reprogramming of both innate and adaptive immune cells to improve host defense mechanisms or organism homeostasis is very attractive and worth of detailed studies. In this context, the article “Immunomodulatory Properties of the Pomegranate Peel Extract…” is very valuable from both scientific and practical point of view. However, I have some comments, which need to be addressed before the acceptance of the manuscript to be published in Pharmaceutics.

Major comments:

  1. Introduction: Because DNA damage and therefore also the mutations you describe as mainly depending on oxidative stress (lines 57-63), the next statements concerning pomegranate effects: “…pomegranate juice has been shown to possess the least antioxidant but the most anti-mutation effects, whereas its methanolic extract showed the most antioxidant but the lowest anti-mutation activity” seems to be in opposition. Thus an explanation sentence is needed.
  2. Introduction: That’s a pity that such important information as immunomodulatory effects of pomegranate (line 89-95) you based only on the review papers [ref. 1, 2, 34, 35], not looking for original works. Add some original studies.
  3. Methods: PBMC were culture and exposed to PoPEx for a very long time (up to 4 days) starting from quite high density (3×105 cells per/well). Did you exchange the medium on fresh containing or not PoPEx during the experiments?
  4. Methods (Subsection 2.4, 2.5) – I expect you prepare the final PoPEx solutions in culture medium, but there is no information about it. Moreover, natural extracts containing polyphenols are usually difficult to dissolve, and such solvents as ethanol / methanol / DMSO are used to prepare stock solutions. What kind of solvent did you used for PoPEx stock solution and what was the final concentration of the solvent in cell culture, because all of them are cytotoxic. Did you prepare an appropriate controls? Complete the information.
  5. Results (line 342-344): “…were seen with all three concentrations (or 50 and/or 12.5 μg/mL)” – First of all you gave only two concentrations in bracket. Moreover, the statement is not entirely compatible: based on Fig. 3 after 24h all three concentrations of PoPEx up-regulated the expression only ATG5 and UVRAG. Improve the description to be more precise.
  6. Results (Subsection 3.4): “The pro-apoptotic concentrations …” – Add which exactly concentration do you mean (moreover, I expect it was only one) to make it easier to follow your further statements: “No significant changes were observed in the mRNA expression for Heme-oxygenase 1 (HMOX1), and thioredoxin (TXN). The expression of several genes (… HMOX1, … and TXN) were upregulated”

Minor comments:

  1. Whole text: Enter a space between value and units, e.g. 100 μg/mL instead of “100μg/mL”
  2. Abstract: The sentence: “…the opposite effect, but inhibited PHA-induced PBMC proliferation and significantly decreased the expression of CD4” is intricate and incomprehensible (requires linguistic correction)
  3. Introduction (line 79): “In the case where” – redundant space before “case”
  4. Whole text: immunostimulator rather than “immunostimulant”
  5. Whole text: T cell (without dash) – Unify in whole text
  6. Materials and methods (line 129): “600 C” - no degree symbol
  7. Materials and methods: “Sigma- Aldrich” (line 161), “25 μg/mL -400 μg/mL” – redundant spaces
  8. Introduction and whole text: “effects of PoPEx on cytotoxicity” – Something may have/not have cytotoxic activity or have/not have a cytotoxic effect on the cells, not “effects on cytotoxicity”. Improve.
  9. Whole text: The cells are cultured not “cultivated”
  10. Results (line 337-338): “Microtubule-associated proteins 1B light chain 3B (MAP1BLCB), than in Fig. 3: MAP1LC3B – Unify the abbreviation
  11. Results (line 359): “…other genes were up-regulated. (Figure 4C). – Redundant dot before bracket
  12. Results (line 430, 482): ”was observed with 100 μg/mL” - was observed at the concentration of
  13. Results (line 450-466) – cytokine production: Subsection should have no 3.7 instead of 2.7; use singular describing cytokine level (e.g. IL-10 level was…, not “IL-10 levels were…”, even at different PoPEx concentrations)
  14. Results (line 482): “TGF-β was up-regulated from 25 μg/mL…” – if “from”, where is “to”? maybe better will be change on: was up-regulated at the concentration of (check whole sentence)
  15. Fig.10: 12.5 μg/mL instead of 12 to make it consistent with other Figures
  16. Discussion (line 645): “was seen only with higher doses” – … at higher doses

Author Response

Major comments:

1.Introduction: Because DNA damage and therefore also the mutations you describe as mainly depending on oxidative stress (lines 57-63), the next statements concerning pomegranate effects: “…pomegranate juice has been shown to possess the least antioxidant but the most anti-mutation effects, whereas its methanolic extract showed the most antioxidant but the lowest anti-mutation activity” seems to be in opposition. Thus an explanation sentence is needed.

We have just cited the explanation from this reference that these opposite effects may be due to variation in the quality and quantity of polyphenols and other bioactive compounds present in different extracts (lines 67-69).

2.Introduction: That’s a pity that such important information as immunomodulatory effects of pomegranate (line 89-95) you based only on the review papers [ref. 1, 2, 34, 35], not looking for original works. Add some original studies.

This sentence was related to the polyphenols from different sources, but anyway three selected references were added (new references [39-41], line 106.)

3.Methods: PBMC were culture and exposed to PoPEx for a very long time (up to 4 days) starting from quite high density (3×105 cells per/well). Did you exchange the medium on fresh containing or not PoPEx during the experiments?

The medium was not changed. This is a universal protocol for PBMC cultures.

4.Methods (Subsection 2.4, 2.5) – I expect you prepare the final PoPEx solutions in culture medium, but there is no information about it. Moreover, natural extracts containing polyphenols are usually difficult to dissolve, and such solvents as ethanol / methanol / DMSO are used to prepare stock solutions. What kind of solvent did you used for PoPEx stock solution and what was the final concentration of the solvent in cell culture, because all of them are cytotoxic. Did you prepare an appropriate controls? Complete the information.

Thanks for this notice. PoPEx was dissolved in DMSO. The final concentration of DMSO in the highest concentration of PoPEx (400 µg/mL) was 0.1% and it was not cytotoxic (see additional explanations, lines 170-172 and 301-302)

  1. Results (line 342-344): “…were seen with all three concentrations (or 50 and/or 12.5 μg/mL)” – First of all you gave only two concentrations in bracket. Moreover, the statement is not entirely compatible: based on Fig. 3 after 24h all three concentrations of PoPEx up-regulated the expression only ATG5 and UVRAG. Improve the description to be more precise.

The description of this part has been improved (see lines 348-352)

  1. Results (Subsection 3.4): “The pro-apoptotic concentrations …” – Add which exactly concentration do you mean (moreover, I expect it was only one) to make it easier to follow your further statements: “No significant changes were observed in the mRNA expression for Heme-oxygenase 1 (HMOX1), and thioredoxin (TXN). The expression of several genes (… HMOX1, … and TXN) were upregulated”

The description of this part has been improved with an indication of applied concentrations (see lines 357, 362, 364)

Minor comments:

1.Whole text: Enter a space between value and units, e.g. 100 μg/mL instead of “100μg/mL”

Corrected in the whole text (marked in red)

  1. Abstract: The sentence: “…the opposite effect, but inhibited PHA-induced PBMC proliferation and significantly decreased the expression of CD4” is intricate and incomprehensible (requires linguistic correction)

Corrected to better explain (lines 23-26)

  1. Introduction (line 79): “In the case where” – redundant space before “case”

Corrected

  1. Whole text: immunostimulator rather than “immunostimulant”

Corrected, although the word immunostimulant is acceptable in the immunological dictionary.

  1. Whole text: T cell (without dash) – Unify in the whole text

Corrected, but in most immunological texts, T-cell is a preferable word (marked in red)

  1. Materials and methods (line 129): “600 C” - no degree symbol

Corrected

  1. Materials and methods: “Sigma- Aldrich” (line 161), “25 μg/mL -400 μg/mL” – redundant spaces

Corrected in the whole text (marked in red)

  1. Introduction and whole text: “effects of PoPEx on cytotoxicity” – Something may have/not have cytotoxic activity or have/not have a cytotoxic effect on the cells, not “effects on cytotoxicity”. Improve.

Corrected (lines 109-114)

  1. Whole text: The cells are cultured not “cultivated”

Corrected

  1. Results (line 337-338): “Microtubule-associated proteins 1B light chain 3B (MAP1BLCB), than in Fig. 3: MAP1LC3B – Unify the abbreviation

Corrected as MAP1LC3B

  1. Results (line 359): “…other genes were up-regulated. (Figure 4C). – Redundant dot before bracket

Corrected

12.Results (line 430, 482): ”was observed with 100 μg/mL” - was observed at the concentration of

Corrected (lines 441,494)

  1. Results (line 450-466) – cytokine production: Subsection should have no 3.7 instead of 2.7; use singular describing cytokine level (e.g. IL-10 level was…, not “IL-10 levels were…”, even at different PoPEx concentrations)

Corrected (lines 462- 478)

  1. Results (line 482): “TGF-β was up-regulated from 25 μg/mL…” – if “from”, where is “to”? maybe better will be change on: was up-regulated at the concentration of (check whole sentence)

Corrected (lines 493- 495)

  1. Fig.10: 12.5 μg/mL instead of 12 to make it consistent with other Figures

Corrected in Fig 10

  1. Discussion (line 645): “was seen only with higher doses” – … at higher doses

Corrected (lines 656)

Round 2

Reviewer 2 Report

I am not satisfied with the Authors reply and have several points of concerns. Apparently, the answer aims to be firm but it results empty and with a lot of discretionality. The manner is a formal and educational matter that does nt impact the level of the science but the lack of appropriate arguments and knowledge of the area of research is a point of major concern when Authors wish to publish in a well qualified journal such as Pharmaceutics.

In detail.

Their answer to my point 1.

The level of endotoxin in PoPEx at the concentration of μg/mL in the complete culture medium was within the accepted limits (0.92 ng/mL), determined by the Limulus amebocyte lysate (LAL) assay

My comment:

The Authors should specify the purchaser of the LAL assay. In addition, what does it mean within the accepted limits ? Accepted from who ? Below accepted limits could be just 0.91 ng/ml. They have to show the precise amount of endotoxin contamination and refer to their own study or literature data that this amount of endotoxin is biologically inhert (even though could be synergistic with their components but that is another storty). Precision is essential in science

Their answer to my previous point 3:

Although our study showed for the first time the complexity of the immunomodulatory activity of PoPEx, translation of these data from in vitro studies to in vivo models and clinical trials requires a detailed analysis of components from PoPEx and their bioactivity. Our current study showed that all three main components (punicalagin, punicalin, and ellagic acid) act synergistically to modulate the immune response. This will be an advantage in understanding the molecular mechanisms involved in these processes".

My comment:

Clearly the answer fails to address the difficulties related to IP of natural products and their drug development and do nt provide any alternative . I suggested for example nanotechnology or "patentable" formulations as alternatives and also suggested a reference (Vallarino et al.). This should be a key issue for patentability but probably they are not familiar with these issues and have not taken proper advantage to study and dissect the reference of Vallarino et al accordingly.

The lack of familiarity of the Authors with basic and common aspects of drug development is also strenghtened by their above phrase that the fact that they have demonstrated that three main components (punicalagin, punicalin, and ellagic acid) act synergisticallycould favour drug development. Ths is incorrect and exactly the contrary. It is very difficult and complex  to develop one drug with 3 components with 3  that act at undefined amounts. 

In addition the statement of "synergism" is scientifically incorrect. They have not demonstrated that the 3 main components act synergistically , to do so they need to work in vitro with  single and combined combinations of the 3 components and demonstrate synergisms or additivity.  What they state is an empiric , say the best, observation not supported by any scientific evidence

Their answer to my previous point 6.

: Some of the suggested references are commented on. Some are not quite relevant, so other ones are incorporated (see lines 784-790).

My comment:

Who has decided which of them are relevant and which not ? Scientific arguments are needed not discretionality. I think it is to say the less surpirsing that in the context of immunomodulatory effects of PoPEX the Authros do nt  discuss specifically the beneficial effects in EAE that I have indicated. Again I feel a lot of self-referentiality not connected  in a homogeneous scientific context.

Their answer to my Point 7:

MIF is not incorporated, because there are so many other pro-inflammatory cytokines that are not investigated. The references for immunoregulatory cytokines IL-35 and IL-37, are added (lines 813-815)

Again I do nt understand the scientific logic of this answer. As many other proinflammatory cytokines are not mentioned , then let us not mention any. Rather, they should quote and discuss that several other proinflammatory cytokines should be studied. 

In addition, since scientific evidence indicates that PoPEX may be important for EAE and multiple sclerosis the Authors could (may be? ) find of interest for themselves and readers of their articles that MIF was not mentioned by chance but for the very simple reason that MIF is the only cytokine for which a  semi-specific inhibitor is used in the clinical setting  (you know, Phase I, II and III studies) of MS. For their curiosity:

Effects of Ibudilast on MRI Measures in the Phase 2 SPRINT-MS Study.

Naismith RT, Bermel RA, Coffey CS, Goodman AD, Fedler J, Kearney M, Klawiter EC, Nakamura K, Narayanan S, Goebel C, Yankey J, Klingner E, Fox RJ; SPRINT-MS investigators.Neurology. 2021 Jan 26;96(4):e491-e500